


# Impacts of droughts and extreme temperature events on gross primary production and ecosystem respiration: a systematic assessment across ecosystems and climate zones

Jannis von Buttlar[1,2], Jakob Zscheischler[1,3], Anja Rammig[4], Sebastian Sippel[1], Markus Reichstein[1,5], Alexander Knohl[2], Martin Jung[1], Olaf Menzer[6], M Altaf Arain[7], Nina Buchmann[3], Alessandro Cescatti[8], Damiano Geinelle[9], Gerard Kiely[10], Beverly E. Law[11], Vincenzo Magliulo[12], Hank Margolis[13], Harry McCaughey[14], Lutz Merbold[3,15], Mirco Migliavacca[1], Leonardo Montagnani[16], Walter Oechel[17,18], Marian Pavelka[19], Matthias Peichl[20], Serge Rambal[21], Antonio Raschi[22], Russell L. Scott[23], Francesco P. Vaccari[22], Eva van Gorsel[24], Andrej Varlagin[25], Georg Wohlfahrt[26], and Miguel D. Mahecha[1,5]

[1]Max Planck Institute for Biogeochemistry, 07745 Jena, Germany
[2]Georg-August-Universität Göttingen, Wilhelmsplatz 1, 37073 Göttingen, Germany
[3]ETH Zürich, Rämistraße 101, 8092 Zürich, Switzerland
[4]TUM School of Life Sciences Weihenstephan, Technische Universität München, 85354 Freising, Germany
[5]German Centre for Integrative Biodiversity Research (iDiv) Halle-Jena-Leipzig, Deutscher Platz 5e, 04103 Leipzig, Germany
[6]University of California Santa Barbara, Santa Barbara CA 93106-3060, U.S.A.
[7]McMaster University, 1280 Main St W, Hamilton, ON L8S 4L8, Canada
[8]JRC, Institute for Environment and Sustainability, TP290 Via E. Fermi, 2749 I-21027 Ispra, Italy
[9]Fondazione Edmund Mach di San Michele all'Adige, Via E. Mach, 1 38010 S. Michele all'Adige, Italy
[10]University College Cork, College Road, Cork T12 YN60, Ireland
[11]Oregon State University, Corvallis OR 97331, USA
[12]CNR, Institute for Mediterranean Forest and Agricultural Systems, via Patacca 85, 80040 Ercolano (Napoli), Italy
[13]Université Laval, 2325, rue de l'Université, Québec G1V 0A6, Canada
[14]Queen's University, 99 University Avenue, Kingston, Ontario, K7L 3N6, Canada
[15]Mazingira Centre, International Livestock Research Institute (ILRI), PO Box 30709, 00100 Nairobi, Kenya
[16]Free University of Bozen-Bolzan, Piazza Università 1, 39100 Bolzano BZ, Italy
[17]Global Change Research Group, San Diego State University, San Diego, CA. USA 92182
[18]Department of Geography, College of Life and Environmental Sciences, University of Exeter, Exeter, UK EX4 4RJ
[19]Global Change Research Institute CAS, Seznam.cz, Bělidla 986/4a, 603 00 Brno, Czech Republic
[20]Department of Forest Ecology & Management, Swedish University of Agricultural Sciences, Skogsmarksgränd, 901 83 Umeå, Sweden
[21]Centre d'Ecologie Fonctionnelle et Evolutive CEFE, 1919, route de Mende 34293 Montpellier 5, France
[22]Istituto di Biometeorologia - Sede di Firenze, Via Giovanni Caproni 8 - I-50145 Firenze, Italy
[23]Southwest Watershed Research Center, 2000 E. Allen Road, Tucson, AZ, 85719, USA
[24]Australian National University, Acton ACT 2601, Canberra, Australia
[25]A.N. Severtsov Institute of Ecology and Evolution, Russian Academy of Sciences, Leninsky pr., 33, Moscow, 119071, Russia
[26]University of Innsbruck, Sternwartestrasse 15, A-6020 Innsbruck, Austria

*Correspondence to:* J. v. Buttlar (jbuttlar@bgc-jena.mpg.de), M. D. Mahecha (mmahecha@bgc-jena.mpg.de)





**Abstract.**

Extreme climatic events, such as droughts and heat stress induce anomalies in ecosystem-atmosphere $CO_2$ fluxes, such as gross primary production (GPP) and ecosystem respiration ($R_{eco}$), and, hence, can change the net ecosystem carbon balance. However, despite our increasing understanding of the underlying mechanisms, the magnitudes of the impacts of different types of extremes on GPP and $R_{eco}$ within and between ecosystems remain poorly predicted. Here we aim to identify the major factors controlling the amplitude of extreme event impacts on GPP, $R_{eco}$, and the resulting net ecosystem production (NEP). We focus on the impacts of heat and drought and their combination. We identified hydrometeorological extreme events in consistently downscaled water availability and temperature measurements over a 30 year time period. We then used FLUXNET eddy-covariance flux measurements to estimate the $CO_2$ flux anomalies during these extreme events across dominant vegetation types and climate zones. Overall, our results indicate that short-term heat extremes increased respiration more strongly than they down-regulated GPP, resulting in a moderate reduction of the ecosystem's carbon sink potential. In the absence of heat stress, droughts tended to have smaller and similarly dampening effects on both GPP and $R_{eco}$, and, hence, often resulted in neutral NEP responses. The combination of drought and heat typically led to a strong decrease in GPP, whereas heat and drought impacts on respiration partially offset each other. Taken together, compound heat and drought events led to the strongest C sink reduction compared to any single-factor extreme. A key insight of this paper, however, is that duration matters most: for heat stress during droughts, the magnitude of impacts systematically increased with duration, whereas under heat stress without drought, the response of $R_{eco}$ over time turned from an initial increase to a down-regulation after about two weeks. This confirms earlier theories that not only the magnitude but also the duration of an extreme event determines its impact. Our study corroborates the results of several local site-level case studies, but as a novelty generalizes these findings at the global scale. Specifically, we find that the different response functions of the two antipodal land-atmosphere fluxes GPP and $R_{eco}$ can also result in increasing NEP during certain extreme conditions. Apparently counterintuitive findings of this kind bear great potential for scrutinizing the mechanisms implemented in state-of-the-art terrestrial biosphere models and provide a benchmark for future model development and testing.

# 1 Introduction

## 1.1 Overview

Extreme climatic events such as heat or drought are key features of Earth's climatic variability (Ghil et al., 2011) that occur on a wide range of time scales (Huybers and Curry, 2006). Extreme climatic events directly propagate into the terrestrial biosphere, thus affecting ecosystem functioning (Reichstein et al., 2013; Frank et al., 2015) and land surface properties (e.g. soil moisture), which in turn triggers ecosystem-atmosphere feedback loops (e.g. Seneviratne et al., 2010; Green et al., 2017). For example, drought in conjunction with severe heat reversed several years of ecosystem carbon sequestration in Europe in 2003 (Ciais et al., 2005), and strong land-atmosphere feedbacks exacerbated the event while it was occurring (Fischer et al., 2007). However, ecosystem impacts of extreme climatic events are often non-linear and interact with concurrent climatic conditions. Additionally, potential impacts can cancel each other out depending on the type and state of the ecosystem and





the magnitude of the climatic event. For instance, extremely warm conditions at the beginning of the growing season during spring 2012 in the contiguous US increased ecosystem carbon uptake, which subsequently compensated for ecosystem carbon losses later during the same year's summer heat and drought. Nonetheless, warm spring conditions and corresponding earlier vegetation activity likely also contributed to exacerbating drought impacts through reduced initial soil moisture at the onset of

summer drought (Wolf et al., 2016). Because extreme climate events have been changing in recent decades with, for example, a general increase in the amount of warm days and the duration of warm spells and the opposite trend for cold days and spells (Sillmann et al., 2013a), and are projected to continue to change (Sillmann et al., 2013b), an understanding of their impacts on ecosystems is crucial. Ideally, this understanding would cover ecological processes that operate at both local and global scales. However, due to non-linear and interacting ecosystem effects of climate extremes, differences in ecosystem responses

across various growing season stages (Wolf et al., 2016), and various ways through which different ecosystem types mediate climatic extremes (e.g. Teuling and Seneviratne, 2011), it currently remains unclear whether a global perspective on ecosystem responses to climate extremes can emerge from local-scale observations alone. Moreover, understanding ecosystem responses to climate extremes is crucial in the context of potentially increasing intensities or frequencies of climatic extremes that could lead to a positive carbon cycle-climate feedback via a reduction in the land carbon sink.

## 1.2   Ecophysiology of photosynthesis and respiration

Gross primary production (GPP, which is carboxylation rate (i.e. true photosynthesis) minus photorespiration, cf. Wohlfahrt and Gu (2015)) is strongly impacted by temperature and water stress (Reichstein et al., 2013). Besides its other main environmental drivers (radiation, humidity (i.e. vapor pressure deficit (VPD)) and $CO_2$ concentration (Leuning, 1990)), temperature directly influences photosynthesis by affecting the kinetics of its two main chemical processes, namely the maximum rates

of carboxylation (i.e. $V_{c,max}$ Farquhar et al. (1980)) and electron transport (i.e. $J_{max}$) (e.g. Medlyn et al., 2002; Sage and Kubien, 2007). Both rates initially increase with rising temperature but decrease above a certain optimum temperature (Bonan, 2008). Leaf (i.e. light) respiration similarly increases with temperature (Leuning, 1990), which additionally reduces GPP. As a result, extremely high temperatures can severely reduce photosynthesis (and, hence, GPP) (Salvucci and Crafts-Brandner, 2004; Allakhverdiev et al., 2008).

Soil water stress impacts photosynthesis (see e.g. van der Molen et al., 2011, for a review) by causing either ecophysiological or structural changes to the plant (Schulze, 1986; Chaves, 1991). For instance, a physiological reduction in photosynthesis can be caused by reductions in enzymatic activity (Chaves et al., 2009; Keenan et al., 2010; van der Molen et al., 2011) or a reduction in mesophyll and stomatal conductance (e.g. Flexas and Medrano, 2002; Bréda et al., 2006). Structural changes reducing photosynthesis include reductions in leaf area and specific leaf area or changes in leaf geometry or orientation (Bréda

et al., 2006; Fisher et al., 2007). Via increased tree mortality, droughts can also severely impact ecosystem level photosynthesis long after the drought event itself (e.g. Bréda et al., 2006; Bigler et al., 2007).

All these responses are highly species dependent, highlighting the need for global cross-site analyses. For example, forest species generally close their stomata much earlier compared to species from grassland or savannah ecosystems, which often keep transpiring until their water storage is depleted (Wolf et al., 2016). In addition, anisohydric plants in general have no





control over their stomata (van der Molen et al., 2011). A soil dependent factor increasing the ecosystem's drought resilience is the rooting depth and the general availability of fine roots (Bréda et al., 2006).

In addition, interactions between heat and drought may affect GPP. For example, drought-induced closing of the stomata and the subsequent reduction in evaporative cooling can further increase heat stress when water stress co-occurs with a high temperature anomaly (De Boeck and Verbeeck, 2011; Bréda et al., 2006). Conversely, high temperature impacts can be alleviated by evaporative cooling as long as enough water for transpiration is available (De Boeck et al., 2010).

Ecosystem respiration ($R_{eco}$) is the sum of autotrophic respiration and the $CO_2$ emissions arising from the heterotrophic decomposition of organic matter in soil (e.g. Law et al., 1999, 2001; Epron et al., 2004)). Like GPP, it is affected by changing soil (and, hence, ambient air) temperatures (Lloyd and Taylor, 1994; Kirschbaum, 1995; Davidson et al., 1998; Kirschbaum, 2006). In addition, the activity of soil microorganisms depends on soil moisture (Orchard and Cook, 1983; Gaumont-Guay et al., 2006; Liu et al., 2009; Epron et al., 2004), and drought conditions directly reduce soil respiration (Jassal et al., 2008). Interactions between the response of these two climate forcings, such as changing temperature dependency due to changing soil water status (e.g. Reichstein et al., 2002, 2007), further complicate the picture.

As described above, both heat and drought affect GPP and $R_{eco}$ in a similar fashion, although the amplitude and onset of this impact may differ. Hence, one important outstanding question involves the impact of climate extremes on the balance of these two fluxes: the net ecosystem production (NEP). Models tend to agree that drought affects GPP more strongly than $R_{eco}$, but their spread is large and predictions for the C balance are uncertain (Zscheischler et al., 2014b). In addition, observational studies on large drought and heat events like the 2003 European heat wave (Ciais et al., 2005; Vetter et al., 2008; Reichstein et al., 2007; Granier et al., 2007) or the 2000-2004 drought in North America (Schwalm et al., 2012) have shown, for example, that drought may cause a much stronger reduction in GPP compared to $R_{eco}$, leading to a reduction in the ecosystem's carbon ($CO_2$) uptake.

However, it is important to understand that the tight coupling between GPP and $R_{eco}$ in most ecosystems (Irvine et al., 2008; Mahecha et al., 2010; Migliavacca et al., 2010; van der Molen et al., 2011; Peichl et al., 2013; Rambal et al., 2014) complicates systematic assessments across sites. For example, heterotrophic respiration is not only a function of the environment, but is also strongly driven by the availability of recently assimilated carbon (Irvine et al., 2005; Granier et al., 2007; Ruehr et al., 2012). Hence, a reduction in photosynthesis may cause a lagged reduction in soil respiration (Law, 2005; Ryan and Law, 2005; Jassal et al., 2012) in the absence of a large labile carbon stock.

### 1.3 Today's opportunities

The majority of studies so far focus on individual sites and predefined extreme events (see Frank et al., 2015, for a review) and only a few have focused on comparisons of extreme event impacts globally across sites and/or across broader regions and different ecosystems (Schwalm et al., 2010, 2012). The La Thuille dataset collected by FLUXNET consists of 252 sites of eddy-covariance flux observations in a standardized way (Baldocchi, 2008). This data provides a basis for a robust assessment of the impacts of climatic extremes on ecosystem $CO_2$ fluxes. The opportunities arising from this observation trove are exemplified in Fig. 1. The figure re-calculates the impacts of the 2003 heat wave on land fluxes as estimated by Ciais et al. (2005) using more



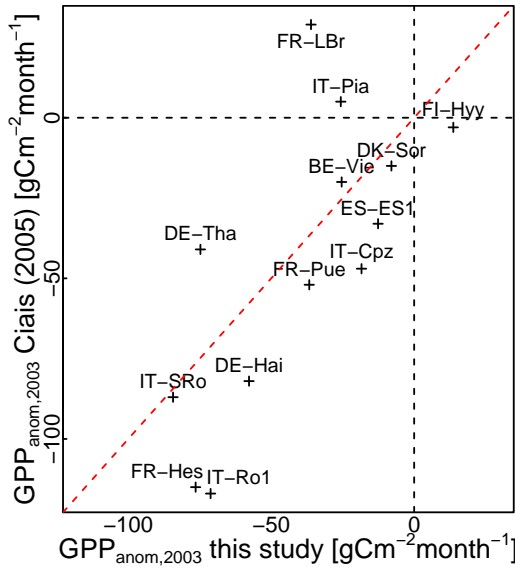

**Figure 1.** Comparison of the gross primary productivity (GPP) reductions during the 2003 European heat wave for several FLUXNET sites. Ciais et al. (2005) quantified this reduction by comparing the 2003 fluxes to the previous year, whereas we are able to use all available site years as a baseline.

reference years based on the data available. The general findings of Ciais et al. (2005), who showed a strong reduction in C uptake, are confirmed, but we now estimate a lower reduction in $CO_2$ uptake when considering more reference years, which is consistent with Vetter et al. (2008), who found a similar pattern using models. Consequently, the length of today's data records and in particular the tremendous work of the numerous networks and initiatives (ref. Sec. A) who collect this data and provide

5 it to the scientific community allows us to update previous quantifications of the $CO_2$ flux impacts of climate extremes.

### 1.4 Objectives of this study

The objectives of this study are threefold: First, we want to exploit the available FLUXNET data to systematically assess if extreme events corroborate our assumptions about ecosystem behavior and to empirically describe the spectrum of extreme responses across the globe. To do so, we extract information about the occurrence of an extreme climatic event directly from the

10 observed data, not by first assuming the occurrence of an extreme event (i.e. by identifying an extreme response of the observed ecosystem). Second, our goal is to develop an extreme event detection framework with a focus not only on the extremeness of the climate forcing but which simultaneously takes into account the resulting extremeness of the ecosystem's response or lack thereof (Smith, 2011; Reichstein et al., 2013). Finally, we aim to bridge the gap between local site level studies and global assessments,which most often are based on models (e.g. Cramer et al., 1999; Friedlingstein et al., 2006) or upscaling studies

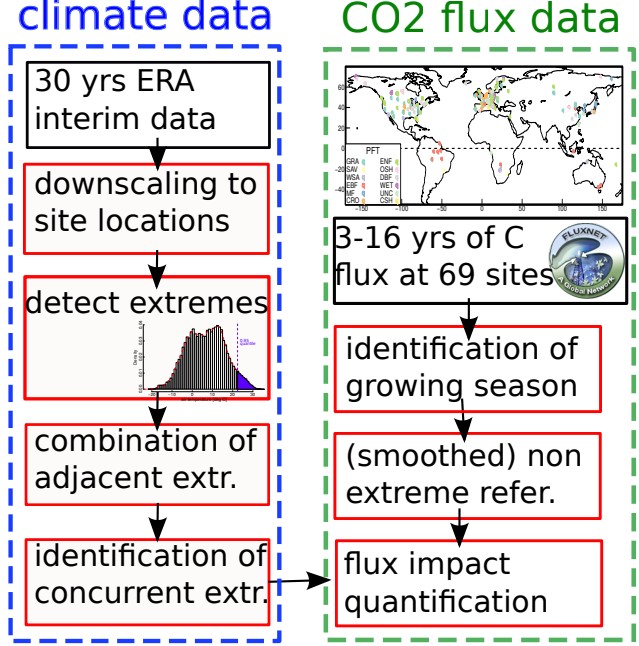

**Figure 2.** Conceptual overview of the different data streams and subsequent steps of our analysis.

(Beer et al., 2010; Jung et al., 2017) by providing some helpful benchmarks for the models and their underlying assumptions (Canadell et al., 2000; Williams et al., 2009).

## 2   Methods

### 2.1   Study concept and overview

5   Our study can be outlined as a three-step process (Fig. 2): first, we use consistently downscaled climate data (Sec. 2.3) to detect climatic extreme events (Sec. 2.4) during the growing season in a set of ecosystems. Second, we compare $CO_2$ fluxes (Sec. 2.2) during these extreme events with reference fluxes during comparable, non-extreme periods to quantify the impact of each extreme event (Sec. 2.5). Third, we use site-specific information like plant functional type (PFT) or eco-climatic zone (Geiger Köppen climate classes) as well as climate extreme characteristics (including type and duration) to systematically

10   assess potential causes of differences between extreme event responses in the different ecosystems.

### 2.2   $CO_2$ flux data

Measurements of $CO_2$ flux and climate parameters collected through a network of measurement sites were used in this study. $CO_2$ fluxes were measured using the eddy-covariance technique (e.g Moncrieff et al., 1997; Baldocchi, 2003). The measured



net carbon flux (i.e. net ecosystem exchange (NEE)) was partitioned into GPP and R$_{eco}$ (Reichstein et al., 2005) at each site. The empirical relationships used by this partitioning scheme assume similar ecophysiological conditions for any given time step (e.g. for one of the extreme events detected here) and a short reference period is used to fit these empirical functions. Environmental stress, however, could also directly impact the processes governing these empirical relationships and hence

the validity of this assumption. To assess whether this could bias our analysis, we also performed all of our calculations using midday NEE as a rough estimate for GPP and averaged nighttime NEE as a proxy for R$_{eco}$. Throughout the rest of the manuscript, we refer to net ecosystem production (NEP) instead of NEE (i.e. $NEP = GPP - R_{eco} = (-1) \cdot NEE$), because NEP is centered on the ecosystem (i.e. positive NEP equals $CO_2$ uptake) and facilitates a more intuitive interpretation together with the component fluxes GPP and R$_{eco}$.

Eddy-covariance measurements are continuously taken at various sites across the globe by individual research teams and are collected and consistently processed by the FLUXNET network (Baldocchi et al., 2001; Baldocchi, 2008, 2014). For this analysis, we used the FLUXNET LaThuile dataset, which consists of a total of 252 sites. We used additional data from the European eddy fluxes database cluster (http://www.europe-fluxdata.eu/) for site-years collected since the creation of the La Thuile dataset in 2007. Both networks consistently filter the submitted data for potential outliers. The half-hourly measurements

supplied by the data providers are consistently gap-filled via marginal distribution sampling (MDS) (Reichstein et al., 2005), i.e. by filling missing values with measurements taken under similar meteorological conditions, and aggregated to daily mean values. For this analysis we used only daily aggregates and excluded data for days with less than 85% original measurements or high confidence gap-filled data.

To be able to compare flux measurements during a potential extreme event with fluxes during non-extreme conditions during

comparable stages of the phenological cycle in other years, we selected 102 sites with time series longer than 3 years. In addition, we removed 4 sites where the correlation between downscaled climate data and measured site meteorology was too low ($R^2 < 0.6$) (Sec. 2.3), and 4 sites where water availability (Sec. 2.3) could not be calculated due to missing data. Finally, we excluded 25 managed and disturbed sites where disturbances such as fire and thinning would have resulted in biases in the calculations of the non-extreme reference data in years before or after the disturbance. This resulted in a subset of 69 sites

(Tab. A3) out of the original 252 LaThuille sites, with a total of 433 site-years of data (i.e. years with data > 75% complete). These sites span eleven plant functional types (PFTs) including grasslands, wetlands and forst type ecosystems (Tab. A1) and all major Geiger Köppen climate zones (Tab. A2) (i.e. first category zones A-E), as well as half of the 24 Geiger Köppen sub-zones (i.e. the secondary categories).

## 2.3   Climate data

To be able to identify extreme events over sufficiently long and consistent time periods for all sites, compared to the much shorter time periods where actual measurements were available, we used downscaled climate data for the extreme event detection. We used daily air temperature, and, for the calculation of the water availability (see below), global radiation and precipitation from ERA-Interim data (Dee et al., 2011) at 0.5°spatial resolution (i.e. the area of 1 pixel $<= (55km)^2$).





Multiple linear regression models of the nearest 9 grid-boxes (i.e. the gridbox with the tower and its direct neighbor pixels) were fitted to FLUXNET site-level meteorology measurements. The resulting models were used to predict site-level values for a time period of 30 years between 1983-2012. The resulting time series were then used to detect climate extremes. The correlation between downscaled and site-level data for air temperature was $R^2 < 0.9$ for nearly 90% of the sites. Sites with $R^2 < 0.6$ ($\approx$5% of the sites, mainly tropical evergreen broad-leaved forests) were removed from the analysis due to the low quality of the downscaling.

To consistently quantify the amount of soil water available to the plant, a water availability index (WAI) was calculated. This index based on the water balance between precipitation and evapotranspiration and was calculated as a simple two-layer bucket model (see supplement 3 in Tramontana et al., 2016, for detailed equations etc.). At each time step, the soil is recharged with water by precipitation up to a maximum value defined by the storage capacity (125 mm). Losses of water by evapotranspiration are taken as the minimum of either potential evapotranspiration or supply limited evapotranspiration. Potential evapotranspiration is calculated based on Priestley and Taylor (1972) using a Priestley-Taylor coefficient of 1.26 and scaled with smoothed fAPAR (from MODIS). Supply limited evapotranspiration is calculated following Teuling et al. (2009), and is simply defined as a fraction (i.e. 0.05, the median of the values determined by Teuling et al. (2009)) of current WAI. Assuming that both water recharge (i.e. precipitation) and water loss (i.e. evapotranspiration) operate from top to bottom, WAI was computed for a simple two-layer model, where the storage capacity of the upper layer was set to 25 mm, and of the lower layer to 100 mm. Only WAI of the lower layer was used in the subsequent analysis and scaled to 0-1 (by dividing by the maximum capacity of 100).

## 2.4 Extreme event detection

*Extreme events* were defined and detected in the following stepwise procedure:

1. identification of single extreme data points (i.e. days) crossing the upper and lower 5th percentile threshold

2. combination of temporally connected single extreme data-points into *extreme events*

3. identification of co-occurring extreme events of different variables to classify *concurrent extremes*.

A percentile based approach (Seneviratne et al., 2012; Zscheischler et al., 2014a) was used to define the upper and lower 5th percentile of the original distribution as extreme (subscript $_{max}$ and $_{min}$). Due to the strong seasonal cycles of air temperature and WAI at most outer tropical sites, this definition resulted in extreme events mainly being detected in summer and winter and represents a means for capturing extreme conditions beyond an actual value with direct physiological meaning.

However, from an ecosystem physiological perspective, an extreme climatic event can also occur outside the maximum or minimum period of the year (e.g. during spring or fall for temperature). To detect such extreme events, air temperature time-series were deseasonalized by subtracting a *mean annual cycle* (MAC) to yield *anomalies*. The MAC was computed as the daily average of all 30 years and smoothed with a 2 week moving-average. The upper (and lower) 5th percentiles of these anomalies were defined as extreme (subscript $_{anom,max}$ and $_{anom,min}$) (Tab. 1 for all extreme event notations used). Such



anomaly extremes were only detected for air temperature, because seasonally varying sensitivity to water availability is not expected.

After the identification of single extreme time steps (i.e. days), contiguous extreme time steps were concatenated into extreme *events*. Additionally, two successive but not contiguous extreme events were subsequently grouped into one long extreme event

if the non-extreme period between them was shorter than 20% of the combined length of the two extreme events together. This prevented short term fluctuations of temperature (WAI did not usually fluctuate so quickly) below the extreme threshold during one long period of high temperature from separating this period into smaller extreme events and allowed for a more realistic assessment of the extreme event duration (see below).

To differentiate between the effects of univariate extremes and the possibility of different impacts of simultaneous extremes

of heat and drought, the following types of extreme events were differentiated: (1) single variable extreme events irrespective of the possible extremeness of other variables (denoted $T/WAI_{min}$), (2) single variable extreme events without other variables being extreme (denoted $T/WAI_{min,\mathbf{s}}$) and (3) concurrent extremes (Seneviratne et al., 2012; Leonard et al., 2014), i.e. coupled extreme events with multiple variables being extreme ($T_{max} + WAI_{min}$) (Tab. 1 for an overview).

Finally all extreme events were described by characteristics such as duration and type (see above) to identify which of these

factors influence the type and magnitude of possible impacts. At this first stage we did not consider several other ecosystem specific important factors which influence the ecosystem's response to climatic extremes such as site history and detailed species composition (e.g. Law, 2014). Such an analysis should be generally possible at future stages (Sec. 3.6), however, the relevant information first has to be gathered across all sites in a standardized and comparable way.

**Table 1.** Overview of different extreme event types and the suffixes denoting them.

| label | extreme type |
| --- | --- |
| $T_{max}/T_{min}$ | temperature maximum/minimum |
| $WAI_{min}$ | water availability minimum (i.e. drought) |
| $T_{anom,max}$ | temperature anomaly maximum |
| $T_{max,s}/WAI_{min,s}$ | temperature/WAI extreme without the other variable being extreme |
| $T_{max} + WAI_{min}$ | concurrent extreme with both temperature and WAI being extreme |

## 2.5  Flux impact calculations

To identify those events that actually have a physiological impact among all the detected climatic extreme events, a consistent quantification of the actual impact on the ecosystem was required.





To do so, differences between the mean of the fluxes during the *extreme event* and comparable *reference periods* were computed (see e.g. Ciais et al., 2005; Schwalm et al., 2012; Van Gorsel et al., 2016; Wolf et al., 2016, for a similar approach). These reference periods were defined to be non-extreme, identical days of the year (DOY) from all other available years. For the reference period, the mean was computed from a moving-average smoothed time series (i.e. 14 day moving-average

filtering computing the median) to minimize the influence of stochastic fluctuations. During the actual extreme event, however, non-smoothed data were used to compute these means.

$$\Delta f = \bar{f}_{extr} - \bar{f}_{ref} = \frac{1}{n} \sum_{k=i}^{i+n-1} f_k - \frac{1}{ny} \sum_{k=j} f_k \tag{1}$$

Here $f$ denotes the respective $CO_2$ flux (NEP, GPP or Reco), $i$ denotes the first day of one particular extreme event of length $n$, $j$ denotes the identical (and not extreme) days of the year (DOY) in all other years, and $y$ is the number of reference years.

As the amplitudes of $R_{eco}$ and GPP differ significantly between highly productive and less productive ecosystems, all analyses were done for original (Eq. 1) and for z transformed time series:

$$\Delta z = \bar{z}_{extr} - \bar{z}_{ref} = \frac{1}{n} \sum_{k=i}^{i+n-1} z_k - \frac{1}{ny} \sum_{k=j} z_k \tag{2}$$

with

$$z_k = \frac{f_k - \bar{f}}{\hat{\sigma}(f)} \tag{3}$$

for all $k$.

Even though extreme events outside the growing season, such as extreme frost periods in winter, can have impacts on the ecosystem's carbon fluxes such impacts would be lagged in many cases (i.e. visible during the following growing season). Because only instantaneous responses were investigated with our framework, it was necessary to exclude such extreme events from the analysis. To identify the growing season, a spline function was used to smooth the time series of GPP. In the first

step, all smoothed values above the 25th percentile were considered to be the growing season. Subsequently, in each year these periods were extended at the beginning and end of the detected period by identifying the first day when the smoothed series dropped below the 5th percentile.

## 3   Results and Discussion

We begin by discussing the different effects of heat and drought on primary production and respiration observed at a global

scale (i.e. averaged over all ecosystems). The different responses to concurrent heat and drought extreme events in contrast to heat or drought only events are highlighted and discussed in Sec. 3.2. The crucial role that the duration of the extreme event plays for the impact is discussed in Sec. 3.3) and the response of different ecosystem types or PFTs that may explain the large



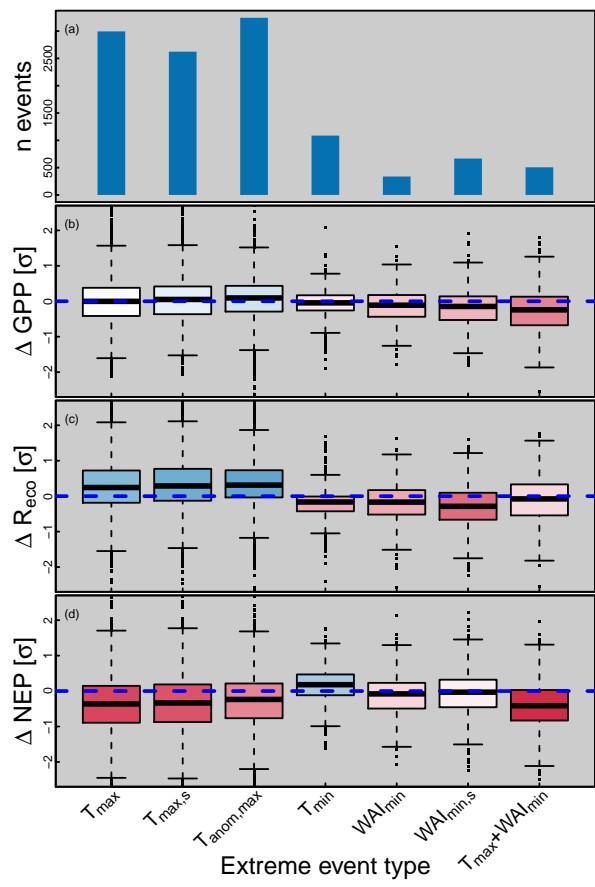

**Figure 3.** Impacts of different extreme event types for a selection of different extreme event types (heat ($T_{max}$), heat only ($T_{max,s}$), temperature anomalies extreme ($T_{anom,s}$), cold ($T_{min}$), drought ($WAI_{min}$), drought only ($WAI_{min,s}$) and combined drought and heat ($T_{max}$+$WAI_{min}$), Tab. 1 for details on all extreme types) on gross primary production (GPP), ecosystem respiration ($R_{eco}$) and net ecosystem production (NEP). Shown are differences between the normalized fluxes (i.e. their z-scores) during a non-extreme reference period and the fluxes during the extreme event ($\Delta z = z_{extr} - z_{ref}$, cf. Eq. 2 and 3 for details). Boxplots are color coded according to the median of the distribution with shades of blue (for positive values, i.e. a flux increase during the extreme) and red (for negative/decreased values).

spread of the impacts in Sec. 3.4. We conclude by discussing strengths and limitations of the approach presented here (Sec. 3.5) and examining future directions (Sec. 3.6).

## 3.1 Contrasting impacts of heat vs. drought on GPP and $R_{eco}$

High temperature extremes without particularly low water availability (i.e. $T_{max}$, $T_{anom,max}$, $T_{max,s}$ and $T_{anom,max,s}$) had

5  only small or virtually zero impacts on GPP (Fig. 3), which is consistent with earlier findings (e.g. for the European heat wave 2003 (Reichstein et al., 2007)). This averaged effect can be partly explained by the specific response of different ecosystem





types (see Sec. 3.4). Heat extremes in general tended to have no or only a small negative impact on observed rates of GPP in most cases. Even though GPP has been shown to have clear temperature optima and decreases at high temperatures due to enzyme inhibition (Bernacchi et al., 2001; Medlyn et al., 2002; Larcher, 2003), such conditions (i.e. temperatures well above 30 °C) are experienced only rarely in the (mostly temperate and Mediterranean) sites investigated. Other studies also confirm

the small impact of heat alone on GPP (De Boeck et al., 2010). Only for very long and pronounced extreme events was a clear negative impact on GPP observed (discussed in detail in Sec. 3.3).

In our analysis, water scarcity events (i.e. $WAI_{min}$ and $WAI_{min,s}$ in general showed a reduction in GPP and $R_{eco}$, which, due to compensation of these component fluxes, led to no discernible changes in NEP on average over the considered FLUXNET sites (Fig. 3). In contrast, events in which low water availability coincided with heat led to a very strong reduction in GPP, but

a lesser reduction in $R_{eco}$ and as a consequence to the strongest reduction in carbon uptake (see Sec. 3.2 for a more detailed discussion). Such a strong effect of droughts (compared to high temperatures alone) on GPP and the generally decreasing effect of drought on GPP is consistent with other studies (e.g. Ciais et al., 2005; Zhao and Running, 2010; Wolf et al., 2013; Zscheischler et al., 2014a, d, c) where water stress directly forces plants to close their stomata to limit transpiration, reducing photosynthesis. Similarly, Jung et al. (2017) found that water availability is a much bigger control on the inter annual variability

of GPP (IAV, which is controlled to a large degree by extreme events) compared to a smaller temperature control on a global level.

In contrast to the small response of GPP to heat, however, $R_{eco}$ generally increased during most high temperature extreme events (Fig. 3). As a consequence, NEP decreased, which represents reduced carbon uptake of the ecosystem. Rising temperatures in general lead to an increase in the microbial degradation of biomass (Mahecha et al., 2010) which explains rising $R_{eco}$

rates during short periods of high temperatures as observed in other studies (Rustad et al., 2001; Wu et al., 2011; Zhao and Running, 2010; Anderson-Teixeira et al., 2011; Van Gorsel et al., 2016). An additional factor could be higher radiation inputs, which result in increased photo-degradation in relatively open non-forest ecosystems.

Compared to temperature, soil respiration as the main component of $R_{eco}$ is regulated much more strongly by soil water availability (Meir et al., 2008). Droughts in general in our study led to a similar reduction in $R_{eco}$ compared to GPP. The reason

for this could be the inhibition of soil microbial processes due to moisture limitation. Additionally, a decrease in GPP also results in a coupling of the two fluxes and, hence, also leads to a reduction in $R_{eco}$ (Högberg et al., 2001; Meir et al., 2008). The compensating effect of drought-induced reductions in both GPP and $R_{eco}$ resulted in small or negligible changes in NEP, which has also been demonstrated at local (e.g. Law, 2005; Meir et al., 2008) and global (Jung et al., 2017) levels.

## 3.2 The differentiated impacts of concurrent heat and drought events on GPP and $R_{eco}$

In contrast to individual events discussed above, concurrent heat and drought extremes ($T_{max}$ + $WAI_{min}$) led to a much stronger reduction in GPP in most cases. On the contrary, $R_{eco}$ was not so strongly (or not at all) reduced. This resulted in the strongest NEP (i.e. C-sink) reduction of any extreme event (Fig. 3).

Several studies have found a lower drought sensitivity of $R_{eco}$ compared to GPP (Ciais et al., 2005; Schwalm et al., 2010, 2012; Rambal et al., 2014; Zscheischler et al., 2014d) whereas we observed comparable or even slightly greater reductions





during drought only (WAI$_{min,s}$) extremes at the global scale. The strong drought extremes investigated in these studies, how-ever, usually coincided with heat extremes and are hence more comparable to our concurrent heat and drought extremes (T$_{max}$+WAI$_{min,s}$) where we also see a nearly negligible mean effect on R$_{eco}$ (compared to GPP).

NEP is the sum of the opposing fluxes of GPP and R$_{eco}$ and hence, the direction and amplitude of its change is always

determined by the sum of the extreme event impacts on the gross fluxes. For heat extremes, the general increase in R$_{eco}$ adds to slight decreases (or no change) of GPP, leading to a generally reduced rate of net carbon uptake. For only drought (and no heat) extremes, the reductions in both GPP and R$_{eco}$ seem to roughly cancel each other out, leading to no strong effects on NEP (again, as a FLUXNET average). However, during the concurrent heat and drought extremes, R$_{eco}$ is less strongly reduced than GPP (and also compared to only drought extremes), leading to strong reductions in net carbon uptake compared

to non-extreme conditions. Part of this effect can be explained by the compensating and opposite effects of heat and drought on R$_{eco}$ (Ciais et al., 2005).

While our analysis confirms a crucial impact of dryness on the individual carbon fluxes GPP and R$_{eco}$, it also shows that drought extreme events in which dryness coincides with T$_{max}$ extremes have a disproportionately large negative impact on the net carbon balance (i.e. compare also Fig. 4 lowest panels on the right side), which is consistent with model results

(Zscheischler et al., 2014b). The combined effect of dryness and heat might be interpreted in a process-oriented way in that dryness acts primarily to reduce GPP, while heat increases R$_{eco}$, thus both leading to a severe reduction in net ecosystem carbon sequestration. Hence, we conclude that indeed an assessment of combinations of extreme climate variables, in particular heat and drought (Zscheischler and Seneviratne, 2017), is crucial for understanding ecosystem impacts (Leonard et al., 2014).

### 3.3  Event duration crucially affects extreme event impacts

Extreme event duration is an important factor that influences ecosystem impacts (Frank et al., 2015). In our study, with in-creasing duration of the extreme climatic event, the impact on GPP generally emerged more clearly (Fig. 4). For T$_{max}$ extreme events there was a threshold at a duration of > 27 days at which GPP strongly decreased by approximately 1-2 $\sigma$. This effect was also visible for R$_{eco}$, albeit less pronounced. However, and somewhat surprisingly, with increasing duration the response in R$_{eco}$ reversed: for short heat events (i.e. a duration of less than 18 days), R$_{eco}$ increased with respect to normal conditions

by up to 2 $\sigma$, whereas for events that last longer than a month, the response of R$_{eco}$ was predominantly negative.

During concurrent T$_{max}$ and WAI$_{min}$ extremes, GPP and R$_{eco}$ were reduced only for extreme events longer than 18 days. For all other extreme types and for the other fluxes, no clear relationship between extreme length and impact was observed (Fig. 4).

The impact of extreme climate events on GPP ranged from a neutral impact (heat lasting less than one week, not coinciding

with dryness) to severe impacts (if temperature extremes persisted for more than a month). The reversal from positive impacts for short durations to negative impacts for long extreme events in the case of R$_{eco}$ might be interpreted as an initial pulse of microbial activity in the soil, which is reduced after some time when the supply limitation of respiration (i.e. GPP effects) kicks in. Hence, these findings highlight that event duration is a critical parameter that might qualitatively affect the directionality of the response, and thus lead to highly non-linear ecosystem responses. These duration effects are often not explicitly considered



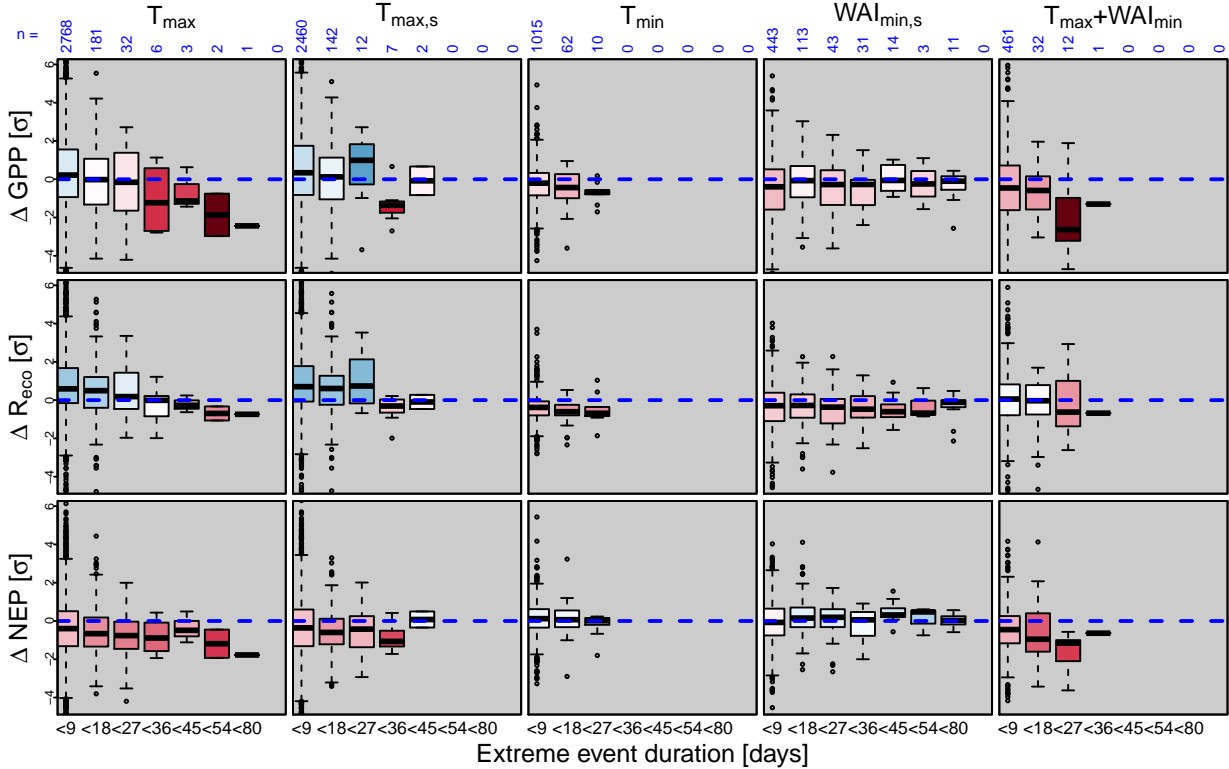

**Figure 4.** Influence of extreme event duration on extreme event impact. Shown are normalized flux differences between extreme events and a reference period ($\Delta z = z_{extr} - z_{ref}$, cf. Eq. 2 and 3 for details) for gross primary production (GPP), ecosystem respiration ($R_{eco}$) and net ecosystem production (NEP) (in rows 1 - 3) for a selection of different extreme event types (in columns 1 - 5: heat ($T_{max}$), heat only ($T_{max,s}$), cold ($T_{min}$), drought only ($WAI_{min,s}$) and combined drought and heat ($T_{max}+WAI_{min}$), Tab. 1 for details on all extreme types).

in the analysis of climate extreme effects on ecosystems (e.g. Ciais et al., 2005; Wolf et al., 2016). Future research should address the question of whether such non-trivial patterns can be reproduced in model simulations.

Most climate extreme indices for temperature consider only relatively short temperature extremes, such as monthly maximum values of temperature or the count or percentage of days that exceed an absolute or relative threshold. Furthermore, currently used climate extreme indices are based on univariate metrics (Sillmann et al., 2013a). Our empirical analysis shows that ecosystem impacts of climate extremes critically depend on the duration of an extreme event and the coincidence of several climate variables. Hence, most critical/negative ecosystem impacts are seen on time-scales of 2-3 weeks to a few months (see also Murray-Tortarolo et al., 2016) and when heat coincides with dryness.




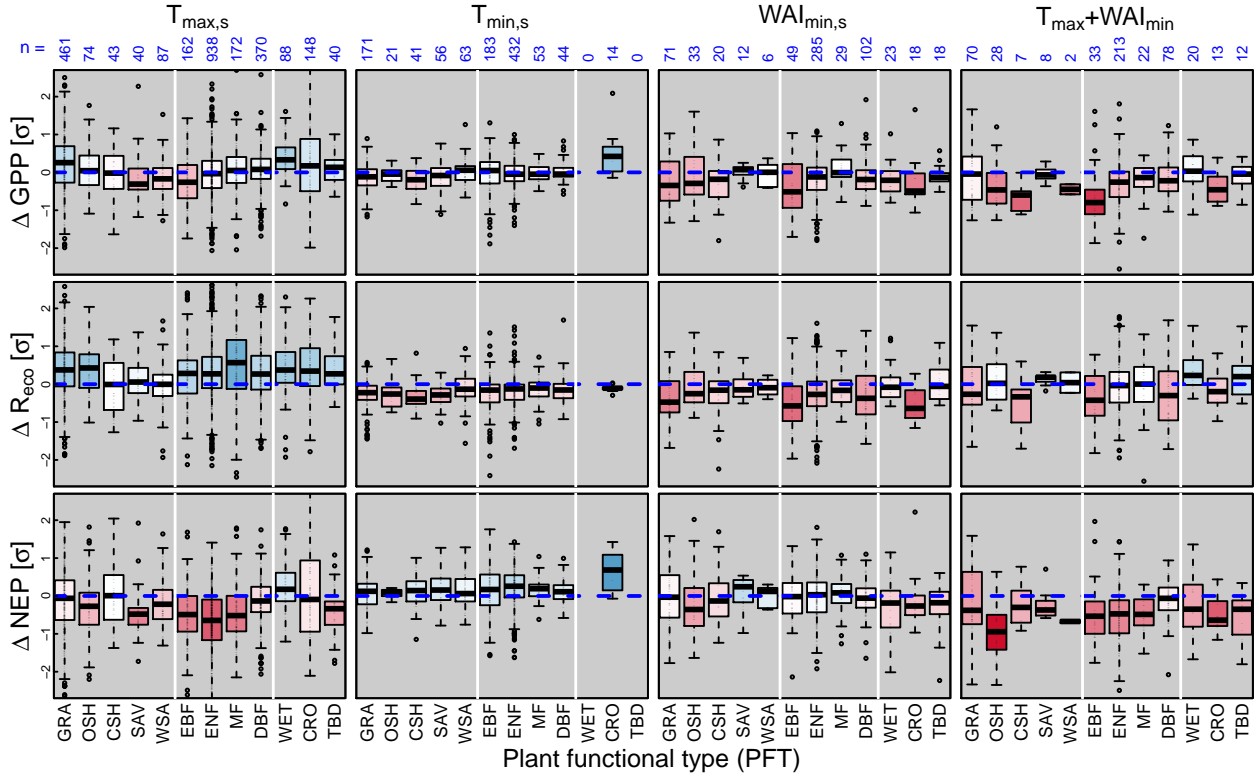

**Figure 5.** Differences between z-transformed $CO_2$ fluxes during extreme events and reference periods for the different extreme event types and the different fluxes ($\Delta z = z_{extr} - z_{ref}$, cf. Eq. 2 and 3 for details) according to the Plant Functional Type (PFT) (Tab. A1 for the acronyms used) of the respective ecosystem (Fig. 6 for a detailed description of the boxplots shown).

### 3.4 Different impacts in different ecosystems

Compared to the differences between the means of the impacts discussed above, the spread of the impacts is rather large (Fig. 3). One reason for this is that differences between ecosystems are hidden by the global (i.e. averaged) focus investigated and discussed above. Fig. 5 shows the extreme event impacts for the different extreme event types separated for the different PFTs,

5   Fig. 6 for different Geiger-Köppen climate classes and Fig. 7 for the combination of the two factors.

The clearest differences between impacts for ecosystems in particular climate zones appeared in the open shrublands (OSH) of the polar climate zone (ET) (Fig. 7). Both GPP and $R_{eco}$ were increased by more than ~1 $\sigma$ during $T_{max}$ extremes (Fig. 6). A stronger increase of $R_{eco}$ led to a slight overall increase of NEP (i.e. a C gain). No drought extremes occurred during the investigated growing seasons in these ecosystems.

10   A similar but smaller (~0.3 $\sigma$) GPP increase during heat extremes occurred in the cold-arid (Bsk), mostly GRA and OSH (Fig. 7), ecosystems (Fig. 6). Here, however, $R_{eco}$ was increased by a similar magnitude, resulting in only a slight increase in NEP.



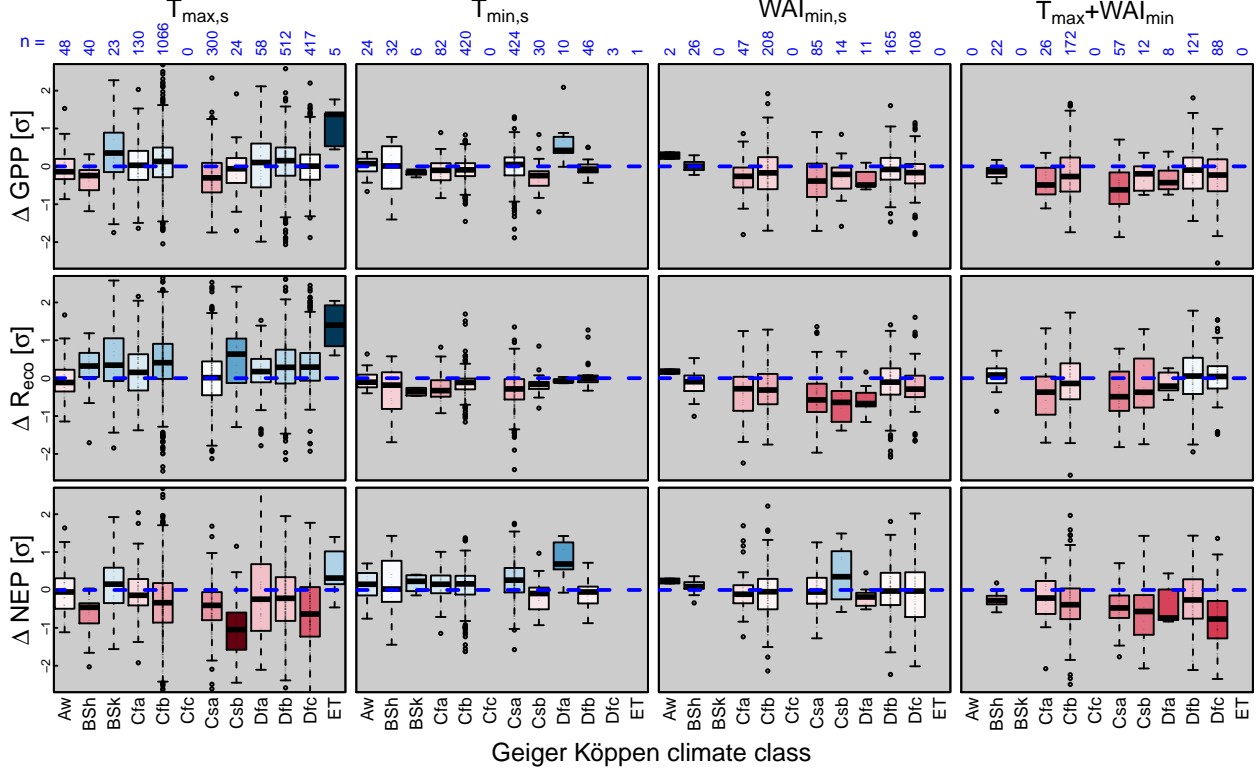

**Figure 6.** Differences between z-transformed $CO_2$ fluxes during extreme events and reference periods ($\Delta z = z_{extr} - z_{ref}$, cf. Eq. 2 and 3 for details) for the different extreme event types (columns 1 - 4: heat alone ($T_{max,s}$), cold alone ($T_{min,s}$), drought alone ($WAI_{min,s}$) and combined heat and drought ($T_{max}+WAI_{min}$) and the different fluxes (rows 1 - 3: gross primary production (GPP), ecosystem respiration ($R_{eco}$) and net ecosystem production (NEP)) according to the Geiger Köppen climate class (Tab. A2 for the acronyms used) of the respective ecosystem.

Again, drought extremes did not occur during the investigated growing seasons. In contrast, in the warm arid (i.e. Bsh climate zone) and exclusively ENF ecosystems, GPP experienced moderate decreases during the $T_{max}$ extremes. In combination with an increase of $R_{eco}$ comparable to the impact in the warm steppe climates (Bsh), this resulted in a general NEP decrease.

The ecosystems in the mostly North American and continental European and Asian 'snow' climate zones (Dfa, Dfb, Dfc)
5  experienced mean increases of $R_{eco}$ during heat extremes of around 0.5 $\sigma$ (Fig. 6). GPP, however, showed almost no changes averaged over the whole Dfc (i.e. cold summer) climate zone during heat extremes but with this being the result of a reduction in its open shrublands (OSH) and opposing increases in the wetlands (WET) of this climate zone (Fig. 7). In hot and warm summer ecosystems of this climate zone (Dfa and Dfb) GPP was slightly increased. As a consequence, this resulted in a relatively strong NEP decrease in Dfc ecosystems, but only a moderate decrease in Dfa and Dfb climates. For drought extremes, however, only





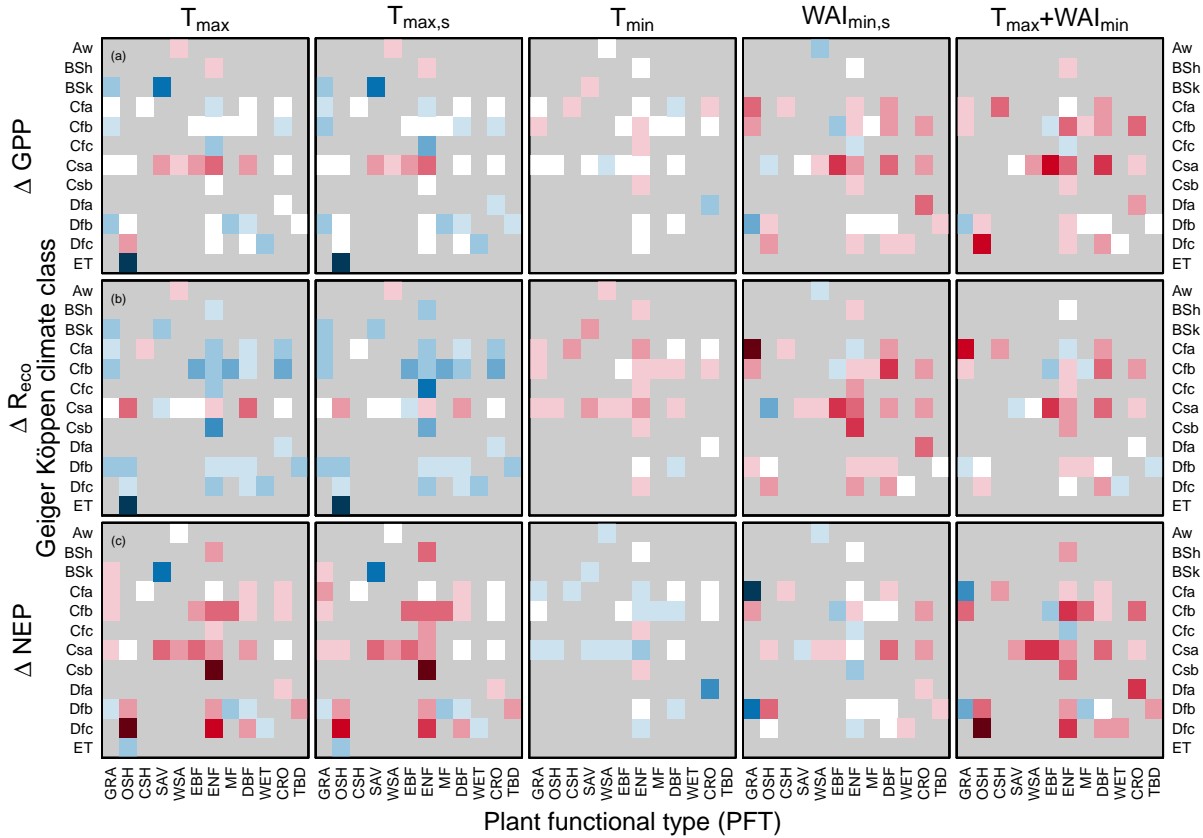

**Figure 7.** Z-transformed flux differences Differences between z-transformed $CO_2$ fluxes during extreme events and reference periods for the different extreme event types and the different fluxes ($\Delta z = z_{extr} - z_{ref}$, cf. Eq. 2 and 3 for details) of the different extreme event types on GPP, $R_{eco}$ and NEP (rows 1 - 3) separated according to plant functional types (PFT) (x axis in each plot, Tab. A1 for the acronyms used) and Geiger Köppen climate class (y axis in each plot, Tab. A2 for the acronyms used) for different types of extreme events (columns 1-5: heat ($T_{max}$), heat alone ($T_{max,s}$), cold ($T_{min,s}$), drought alone ($WAI_{min,s}$) and combined heat and drought ($T_{max}+WAI_{min}$)). Shades of red indicate reductions of different size in the respective fluxes, shades of blue increases. Refer to figures 5 and 6 for a visualization or quantification of the actual magnitude of these impacts.

the summer hot Dfa cropland (CRO) ecosystems showed reductions in $R_{eco}$ and, to a lesser extent, in GPP during drought extremes.

Temperate and summer hot and dry (Csa, mainly Mediterranean) ecosystems experienced the strongest GPP reductions (0.3 $\sigma$), with particularly strong impacts in the forest and savannah ecosystem compared to grasslands and open shrublands (Fig. 7), during heat extremes (Fig. 6), whereas $R_{eco}$ in general was not impacted, resulting in a NEP decrease (Fig. 6) during heat extremes. During drought periods, these Csa sites were among the ecosystems with the strongest reductions in $R_{eco}$ for all forest and savannah ecosystems but not the open shrublands which experienced increases in respiration (Fig. 7), and to a lesser




extent in GPP. In contrast, temperate summer dry ecosystems with only warm summers (Csb) did not experience such strong reductions in GPP and even increases of R$_{eco}$ during heat extremes and a smaller decrease of GPP during drought extremes (compared to Csa). Most other ecosystems in humid temperate climate zones (Cfa and Cfb) showed impacts consistent with the general patterns (i.e slight GPP and stronger R$_{eco}$ increases during heat extremes, a reduction in both fluxes during drought

and a smaller reduction in R$_{eco}$ during concurrent heat and drought).

The few equatorial winter dry (Aw) woody savanna ecosystems under investigation experienced slight reductions in GPP during heat extremes. They were one of the few climate zones where R$_{eco}$ was slightly reduced during T$_{max}$ extremes. Due to the few sites and short time series, drought extremes did not occur here often enough to reliably investigate their impacts.

Whether temperature or water availability governs an ecosystem's response to extreme events is mainly dependent on

whether the ecosystem is located in a temperature or water-limited environment (Nemani et al., 2003). This explains the strong increases of both GPP and R$_{eco}$ during high temperature extremes in the open shrublands of the temperature-limited polar ET climate zone compared to all other climatic zones (Fig. 8). Similar results have been found by Wu et al. (2011). In addition, the detected extreme events are at relatively low temperatures below 20 °C, which are probably well below a possible heat stress for the affected plants and still in the range where increasing temperatures increase both GPP rates and the decomposition

processes which govern R$_{eco}$. An additional factor could have been the increased sunlight during the extreme events (which may have caused the heat extreme in the first place) in these energy-limited regions.

Temperature extremes at sites of the arid steppe climates (BSh and BSk) have comparatively small impacts, probably because most heat extremes occur during dry periods with very low biological activity (Fig. 8). For one BSk site, however, the period of high temperatures and high fluxes coincides with GPP increases during these extreme events, causing the general mean GPP

increase in this climate class compared to the BSh sites.

Temperate and summer hot and dry (Csa, mainly Mediterranean) ecosystems experienced the strongest GPP reductions (0.3 σ) during heat extremes, whereas R$_{eco}$ in general was not impacted, resulting in a moderate NEP decreases (Fig. 6) during heat extremes. During drought periods, these sites were among the ecosystems with the strongest reductions in R$_{eco}$ and to a lesser extent GPP. In contrast, temperate summer dry ecosystems with warm summers (Csb) did not experience such strong reductions

in GPP, and even experienced increases of R$_{eco}$ during heat extremes and a smaller decrease of GPP during drought extremes (compared to Csa). Most other ecosystems in humid temperate climate zones (Cfa and Cfb) showed impacts consistent with the general patterns (i.e slight GPP and stronger R$_{eco}$ increases during heat extremes, a reduction in both fluxes during drought and a smaller reduction in R$_{eco}$ during concurrent heat and drought (Schwalm et al., 2010; Wolf et al., 2016).

For the one available tropical Aw site, very small seasonal temperature changes between ≈ 30°C and 32°C are observed

(Fig. 8). As a result, our extreme detection framework detects all extreme events during the slightly hotter rainy season at the beginning and end of the year (in the Southern Hemisphere). Still, such small temperature differences are unlikely to cause visible physiological impacts, which is demonstrated by the nearly non-existent mean impact on GPP, R$_{eco}$ and NEP in this climate zone. However, station density in tropical ecosystems is very low compared to temperate Northern European or North American sites so this may also be a consequence of the small amount of extreme events detected.





**Figure 8.** Yearly cycles for climatic forcing variables (air temperature and the water availability index (WAI)) and carbon fluxes (gross primary production (GPP), ecosystem respiration ($R_{eco}$) and net ecosystem production (NEP)) for one example site for each different climatic region (i.e. Geiger Köppen climate class, Tab.A2). Shown are one example year (black dots) with various detected extreme events (red dots). Grey dots represent all reference data from other years. Colored backgrounds indicate the different extreme events detected in the example year.

## 3.5 Opportunities and limitations of our approach

The approach presented in this paper is based on a global, empirical characterization of the impacts of climate extremes on ecosystem-atmosphere carbon fluxes, which has several advantages but also limitatons for addressing global ecological questions. Classical extreme event research has often focused on events where the response was already known *a priori* to




be strong and has possibly neglected several comparable climatic periods with similar conditions but with smaller or even opposite impacts. In contrast, all periods are included in our analysis because we did not select our extreme events *a priori*. Our results show that comparable extreme events can lead to contrasting impacts, which depend on ecosystem type or extreme event timing.

In addition, this research is one of the few global and cross-site/ecosystem investigations of extreme climate impacts on (measured) $CO_2$ fluxes. We try to extend the sometimes limiting (but still valuable) focus on particular sites and compare such responses globally. This allows for a holistic picture with which such local site observations can be compared.

     Our global results highlight the importance of drought events for the ecosystem carbon cycle. Hence, a reliable estimate of water availability is crucial for the identification of climatic extreme events. As soil water measurements at FLUXNET
sites differ strongly between sites in quality, depth and duration, we chose to use the modeled WAI for better between-site comparability and consistency (e.g. Tramontana et al., 2016). Even though we see responses of the fluxes to decreasing WAI, the detailed investigation of individual drought events (e.g. the 2003 heat wave: Fig. 1) highlighted the possible sudden decrease of the fluxes to gradual changes in WAI, emphasizing the need for a reliable estimate of WAI. At this stage, WAI was not optimized for the individual sites and represents a purely hydrometeorological variable rather than a direct measure of ecosystem-specific
water stress.

     We applied the 95th (or 5th) percentile threshold to define extreme events throughout our study to allow for a comparable extreme definition for all ecosystems. Importantly, this approach has as few *a priori* assumptions as possible (compared to identifying extreme events via somehow "subjective" expert knowledge or by identifying extreme events using extreme responses) allowed us to thoroughly test such assumptions. However, this approach also has some limitations. First, enforcing
this extreme definition always leads to a fixed number (i.e. 5%) of "extreme" days per site. For long enough and strongly varying time series, this approach yields actual extreme events. However, for shorter time series or sites with weakly varying climate (e.g. tropical sites), this method may lead to a *false positive* extreme event identification of non-extreme conditions. The WAI extreme detection is probably more strongly affected by this problem. For the rather smooth time series with long periods of low and only slightly varying WAI at several sites (see for example the IT Ro1 WAI time series of 2003 in Fig. 1) this
approach probably led to rather arbitrary breaks between extreme and non-extreme timespans caused by only very small WAI differences. A more flexible data-driven approach to determine site-specific extreme thresholds may be helpful for alleviating this problem in future approaches. For WAI in particular, an ecosystem and soil type specific threshold may lead to improved results. Finally, future approaches should take additional extreme strength indicators like amplitude or occurrence into account when defining the extreme threshold. One also has to note that FLUXNET sites are not necessarily well placed to capture
extreme events (Mahecha et al., 2017).

     We used changes of the $CO_2$ fluxes to quantify the impact of the extreme events. Such changes, however, can only be defined relative to an undisturbed reference period. Due to the strong seasonal cycles at many of the investigated sites, we used fluxes from other years but identical periods (in the year) as these reference values. However, shifts of the phenological cycle between years could bias these reference values, especially during stages of steep phenological changes at the beginning and end of the
growing season. We used smoothed data from multiple years to attenuate this effect. A possibly promising future improvement




would be to synchronize each yearly cycle with a reference by shifting it in time until a maximum agreement is reached. For short extreme events, the impact could alternatively be calculated with regard to the fluxes before and or after the extreme.

### 3.6 Future directions

In addition to the methodological modifications and improvements outlined above (Sec.3.5) there are several promising
methodological extensions and possibilities.

For strongly fluctuating time series such as air temperature, our method of defining individual days as extreme and subsequently joining them into concurrent extreme events often resulted in the identification of several successive but interrupted events. These were then analyzed and treated independently, which may neglect their cumulative impact (e.g. Bréda et al., 2006; Granier et al., 2007) on the ecosystem. We alleviated this effect by joining large extreme events with small gaps in be-
tween, but our choice of when to join the extreme events and when to treat them separately was rather *ad-hoc*. Such problems could be solved by applying a moving-window based approach when detecting the extreme events, which takes into account the "extremeness" of a defined period before each individual day. In particular, this approach could improve the results for the multivariate extreme events where the fluctuations in temperature led to many small, fragmented extreme events.

In addition, our method for defining multivariate extreme events is (intentionally) simple and suffers from some restrictions.
By independently identifying extreme events in each climate forcing (i.e. temperature and WAI), we may miss out potentially differing impact thresholds in situations when both forcings are extreme. A true multivariate extreme detection methodology, possibly also including other variables such as vapor pressure deficit or radiation, could overcome this limitation (e.g. Seneviratne et al., 2012; Leonard et al., 2014; Flach et al., 2016; Zscheischler and Seneviratne, 2017).

One important aspect of extreme event impacts on ecosystems not covered by the approach presented here are *lagged* or
*carry-over* (i.e. memory) effects (e.g. Krishnan et al., 2006; Bréda et al., 2006; Bigler et al., 2007; Arnone Iii et al., 2008; Thomas et al., 2009). These are impacts which persist even after the end of the actual extreme or occur only after the event or during subsequent growing seasons. In addition, extreme events outside of the growing season (i.e. frost events during winter) are not investigated here. We chose to focus on instantaneous effects and neglect such lagged aspects because only the direct and unambiguous connection of possible impacts to one unique extreme event ensured a large enough sample size to apply the
assumption-free approach and test all possible extreme event sizes and types for impacts. However, focusing on a subset of long and pronounced extreme events, an identical approach could be used to assess non-instantaneous effects. An additional interesting aspect would be to examine the effect of the size of the time span between the extreme event onset and the flux response for $R_{eco}$ and GPP (i.e. the size of the "lag") and a possible difference between the two fluxes (e.g. Zscheischler et al., 2014b).





## 4 Conclusions

In this study we evaluated and corroborated the current understanding and hypotheses about the response of ecosystem $CO_2$ fluxes to extreme climatic events. We aimed for a strictly data-driven and assumption-free approach that takes into account both the 'extremeness' of the climate forcing and that of the response.

Our approach first defines extreme values in the climate data (i.e. the highest and lowest 5%) to detect extreme events of varying length and then calculates the difference between $CO_2$ fluxes during these events compared to non-extreme reference periods.

    We found that periods of dryness (without extraordinary heat) reduce both GPP and $R_{eco}$, which led to a relatively neutral across-site impact in net ecosystem carbon sequestration. In contrast, heat without dryness increased $R_{eco}$ but did not consis-

tently affect GPP (partly because of differentiated effects across ecosystem types and event duration), which overall led to a reduction of NEP. If heat coincided with drought, these events strongly reduced GPP, but yielded smaller reductions in $R_{eco}$, which led to strong reductions in NEP. A crucial contributing factor to these differentiated impacts was the duration of the respective climate extreme events: for instance, under heat extremes, $R_{eco}$ initially increased (for the first 18 days on average) relative to non-extreme conditions, but decreased for longer events, presumably due to a reduction in GPP and thus in soil

carbon pools for long heat events.

    Similar extreme events at similar sites in several cases led to decreases but also to increases of $CO_2$ fluxes, i.e. a large spread remained in the data. These different responses could be partly linked to ecosystem-specific factors. For example, boreal ecosystems experienced strong increases in GPP and $R_{eco}$ during heat extremes compared to smaller changes in most other ecosystems, whereas Mediterranean summer dry ecosystems showed particularly strong flux decreases during drought

extremes. However, uncertainties and somewhat diverging impacts still remain unexplained after accounting for ecosystem type, climate zone, and event duration.

    The framework proposed here forms a suitable basis for several promising modifications and more in-depth analyses in the future. We plan to address these open questions by improving the extreme detection methodology and performing an in-depth investigation of several additional aspects. As responses to heat and drought also influence the exchange of water and, hence,

the fluxes of water and energy (e.g. Bonan, 2015) and such fluxes are also measured by the eddy covariance technique (i.e. their net balance), we plan to conduct a similar analysis with these fluxes, as has been done for individual events (e.g. Teuling et al., 2010). Other important aspects to include in future studies are the timing of the extreme during the growing season, which can significantly influence the response (Schwalm et al., 2010; De Boeck and Verbeeck, 2011; Wolf et al., 2013). Eddy covariance measurements continue to be collected so for several FLUXNET sites increasingly long time series are becoming available.

Hence we are looking forward to future data releases and to the possibility of extreme event detection using the measured data directly, without the constraints and possible biases of the downscaling, which highlights the crucial importance of continuous long-term measurements for meaningful ecosystem and climate research.



## Appendix A: Appendix A

**Table A1.** Description of the Plant Functional Type (PFT) classes of the ecosystems investigated in this study (according to the IGBP vegetation classification scheme).

| Class | Name | Detailed Description |
|---|---|---|
| CRO | Croplands | temporary crops |
| CSH | Closed Shrublands | woody shrub vegetation |
| DBF | Decid. Broadleaf For. | seasonal broadleaf trees |
| EBF | Evergr. Broadleaf Forests | evergreen broadleaf trees |
| ENF | Evergr. Needleleaf Forests | evergreen needleleaf trees |
| GRA | Grasslands | herbaceous types (tree and shrub cover < 10%) |
| MF | Mixed Forests | mixture of all tree types |
| OSH | Open Shrublands | woody vegetation (cover between 10-60%) |
| SAV | Savannas | herbaceous and other understory systems (woodland between 10-30%). |
| WET | Permanent Wetlands | permanent mixture of water and herbaceous or woody vegetation |
| WSA | Woody Savannas | herbaceous/other understory veg. (woodland between 30-60%,) |





**Table A2.** Description of Geiger-Köppen climate classes after Kottek et al. (2006) defined by temperature (T) and precipitation (P) (with $P_{th}$ being a dryness threshold and $_s$ and $_w$ denoting summer and winter values respectively (c.f. Kottek et al., 2006, for details)

| Class | Description | Characteristics |
|---|---|---|
| A | equat. cl. | $T_{min} \geq 18°C$ |
| Af | equat. full humid rainforest | $P_{min} > 60mm$ |
| Am | equat. monsoon | $P_{ann} \geq 25(100mm - P_{min})$ |
| As | equat. Savannah + dry sum. | $P_{min} < 60mm$ in summer |
| Aw | equat. Savannah + dry win. | $P_{min} < 60mm$ in winter |
| B | Arid cl. | $P_{ann} < 10\,P_{th}$ |
| BS | Steppe cl. | $P_{ann} > 5\,P_{th}$ |
| BW | Desert cl. | $P_{ann} \leq 5\,P_{th}$ |
| C | Warm temp. cl. | $-3°C < T_{min} < 18°C$ |
| Cs | Warm temp. cl. + dry sum. | $P_{min,s} < P_{min}$; $P_{max,w} > 3P_{min,s}$ and $P_{min,s} < 40mm$ |
| Cw | Warm temp. cl. + dry win. | $P_{min,w} < P_{min,s}$ and $P_{max,s} > 10P_{min,w}$ |
| Cf | Warm temp. fully humid | cl. neither Cs nor Cw |
| D | Snow cl. | $T_{min} \leq -3°C$ |
| Ds | Snow cl. + dry summer | $P_{min,s} < P_{min,w}$; $P_{max,w} > 3P_{min,s}$ and $P_{min,s} < 40mm$ |
| Dw | Snow cl. + dry winter | $P_{min,w} < P_{min,s}$ and $P_{max,s} > 10P_{min,w}$ |
| Df | Snow cl., fully humid | neither Ds nor Dw |
| E | Polar cl. | $T_{max} \leq 10°C$ |
| EF | Tundra cl. | $0°C \leq T_{max} < 10°C$ |
| ET | Frost cl. | $T_{max} <= 0°C$ |

third letter:

| | | |
|---|---|---|
| h | Hot steppe / desert | $T_{ann} \geq 18°C$ |
| k | Cold steppe /desert | $T_{ann} < 18°C$ |
| a | Hot summer | $T_{max} \geq 22°C$ |
| b | Warm summer | not (a) + $> 4T_{mon} \geq 10°C$ |
| c | Cool summer and cold winter | not (b) and $T_{min} > -38°C$ |
| d | extremely continental | like (c) but $T_{min} \leq -38°C$ |





Table A3: List of FLUXNET sites analysis with their code, name, country, geographical location, Geiger Köppen climate (GKC) class, plant functional type (PFT) and the measurement time periods of the data used.

| Code | Name | Country | Latitude | Longitude | PFT | KGC | Site-years | Reference |
|------|------|---------|----------|-----------|-----|-----|------------|-----------|
| AU-How | Howard Springs | Australia | -12.49 | 131.15 | WSA | Aw | 2001-2006 | Beringer (2003) |
| AU-Tum | Tumbarumba | Australia | -35.66 | 148.15 | EBF | Cfb | 2001-2006 | Finnigan and Leuning (2000) |
| BE-Bra | Brasschaat (De Inslag Forest) | Belgium | 51.31 | 4.52 | MF | Cfb | 1999-2009 | Carrara et al. (2004) |
| BE-Lon | Lonzee | Belgium | 50.55 | 4.74 | CRO | Cfb | 2004-2010 | Moureaux et al. (2006) |
| BE-Vie | Vielsalm | Belgium | 50.31 | 6.00 | MF | Cfb | 1996-2011 | Aubinet et al. (2001) |
| CA-Ca1 | Campbell River - Mature Forest Site | Canada | 49.87 | -125.33 | ENF | Cfb | 1997-2005 | Morgenstern et al. (2004) |
| CA-Let | Lethbridge | Canada | 49.71 | -112.94 | GRA | Dfb | 1998-2005 | Flanagan et al. (2002) |
| CA-Man | BOREAS NSA - Old Black Spruce | Canada | 55.88 | -98.48 | ENF | Dfc | 1994-2003 | Lafleur et al. (2003) |
| CA-Mer | Eastern Peatland- Mer Bleue | Canada | 45.41 | -75.52 | WET | Dfb | 1998-2005 | Lafleur et al. (2003) |
| CA-NS2 | UCI-1930 burn site | Canada | 55.91 | -98.52 | ENF | Dfc | 2001-2005 | Goulden et al. (2006) |
| CA-NS3 | UCI-1964 burn site | Canada | 55.91 | -98.38 | ENF | Dfc | 2001-2005 | Goulden et al. (2006) |
| CA-NS6 | UCI-1989 burn site | Canada | 55.92 | -98.96 | OSH | Dfc | 2001-2005 | Goulden et al. (2006) |
| CA-Oas | Sask.- SSA Old Aspen | Canada | 53.63 | -106.20 | DBF | Dfc | 1997-2005 | Black et al. (1996) |
| CA-Obs | Sask.- SSA Old Black Spruce | Canada | 53.99 | -105.12 | ENF | Dfc | 1999-2005 | Jarvis et al. (1997) |
| CA-Ojp | Sask.- SSA Old Jack Pine | Canada | 53.92 | -104.69 | ENF | Dfc | 1999-2005 | Baldocchi et al. (1997) |
| CA-Qfo | Quebec Mature Boreal Forest Site | Canada | 49.69 | -74.34 | ENF | Dfc | 2003-2006 | Bergeron et al. (2007) |
| CH-Oe1 | Oensingen1 grass | Switzerland | 47.29 | 7.73 | GRA | Cfb | 2002-2008 | Ammann et al. (2007) |
| CZ-BK1 | Bily Kriz- Beskidy Mountains | Czech Republic | 49.50 | 18.54 | ENF | Dfb | 2000-2012 | Havránková and Sedlák (2004) |
| CZ-BK2 | Bily Kriz- grassland | Czech Republic | 49.50 | 18.54 | GRA | Dfb | 2004-2011 | Marek et al. (2011) |
| DE-Geb | Gebesee | Germany | 51.10 | 10.91 | CRO | Cfb | 2002-2008 | Anthoni et al. (2004) |
| DE-Hai | Hainich | Germany | 51.08 | 10.45 | DBF | Cfb | 2000-2007 | Knohl et al. (2003) |
| DE-Wet | Wetzstein | Germany | 50.45 | 11.46 | ENF | Cfb | 2005-2008 | Anthoni et al. (2004) |
| DK-Sor | Soroe- LilleBogeskov | Denmark | 55.49 | 11.65 | DBF | Cfb | 1996-2009 | Pilegaard et al. (2001) |
| ES-ES1 | El Saler | Spain | 39.35 | -0.32 | ENF | Csa | 1999-2006 | Reichstein et al. (2005) |
| ES-LMa | Las Majadas del Tietar | Spain | 39.94 | -5.77 | SAV | Csa | 2004-2011 | Perez-Priego et al. (2017) |
| FI-Kaa | Kaamanen wetland | Finland | 69.14 | 27.30 | WET | Dfc | 2000-2008 | Aurela et al. (2004) |
| FI-Sod | Sodankyla | Finland | 67.36 | 26.64 | ENF | Dfc | 2000-2008 | Thum et al. (2007) |
| FR-Fon | Fontainebleau | France | 48.48 | 2.78 | DBF | Cfb | 2005-2008 | Michelot et al. (2011) |
| FR-LBr | Le Bray (after 6/28/1998) | France | 44.72 | -0.77 | ENF | Cfb | 1996-2008 | Berbigier et al. (2001) |
| FR-Lq1 | Laqueuille | France | 45.64 | 2.74 | GRA | Cfb | 2004-2010 | Allard et al. (2007) |
| FR-Lq2 | Laqueuille extensive | France | 45.64 | 2.74 | GRA | Cfb | 2004-2010 | Allard et al. (2007) |
| FR-Pue | Puechabon | France | 43.74 | 3.60 | EBF | Csa | 2000-2011 | Rambal et al. (2004) |
| HU-Bug | Bugacpuszta | Hungary | 46.69 | 19.60 | GRA | Cfb | 2002-2008 | Nagy et al. (2007) |
| HU-Mat | Matra | Hungary | 47.85 | 19.73 | GRA | Cfb | 2004-2008 | Nagy et al. (2007) |
| IE-Dri | Dripsey | Ireland | 51.99 | -8.75 | GRA | Cfb | 2003-2007 | Peichl et al. (2011) |
| IL-Yat | Yatir | Israel | 31.34 | 35.05 | ENF | BSh | 2001-2006 | Grünzweig et al. (2003) |
| IT-Amp | Amplero | Italy | 41.90 | 13.61 | GRA | Cfa | 2002-2008 | Wohlfahrt et al. (2008) |
| IT-Cpz | Castelporziano | Italy | 41.71 | 12.38 | EBF | Csa | 1997-2008 | Tirone et al. (2003) |
| IT-Lav | Lavarone (after 3/2002) | Italy | 45.96 | 11.28 | ENF | Cfb | 2000-2012 | Cescatti and Marcolla (2004) |
| IT-LMa | La Mandria | Italy | 45.58 | 7.15 | GRA | Cfb | 2003-2009 | Maselli et al. (2006) |
| IT-MBo | Monte Bondone | Italy | 46.02 | 11.05 | GRA | Cfb | 2003-2012 | Marcolla and Cescatti (2005) |
| IT-Non | Nonantola | Italy | 44.69 | 11.09 | DBF | Cfa | 2001-2008 | Nardino et al. (2002) |
| IT-Pia | Island of Pianosa | Italy | 42.58 | 10.08 | OSH | Csa | 2002-2006 | Vaccari et al. (2012) |
| IT-SRo | San Rossore | Italy | 43.73 | 10.28 | ENF | Csa | 1999-2010 | Chiesi et al. (2005) |
| JP-Tak | Takayama | Japan | 36.15 | 137.42 | DBF | Dfb | 1999-2004 | Yamamoto et al. (1999) |

*Continued on next page*




Table A3 – *Continued from previous page*

| Code | Name | Country | Latitude | Longitude | PFT | KGC | Site-years | Reference |
|------|------|---------|----------|-----------|-----|-----|-----------|-----------|
| JP-Tom | Tomakomai National Forest | Japan | 42.74 | 141.51 | MF | Dfb | 2001-2003 | Hirano et al. (2003) |
| NL-Hor | Horstermeer | Netherlands | 52.03 | 5.07 | GRA | Cfb | 2004-2010 | Hendriks et al. (2007) |
| NL-Loo | Loobos | Netherlands | 52.17 | 5.74 | ENF | Cfb | 1996-2012 | Dolman et al. (2002) |
| PT-Esp | Espirra | Portugal | 38.64 | -8.60 | EBF | Csa | 2002-2008 | Rodrigues et al. (2011) |
| PT-Mi2 | Mitra IV Tojal | Portugal | 38.48 | -8.02 | GRA | Csa | 2004-2008 | Pereira et al. (2007) |
| RU-Fyo | Fyodorovskoye wet spruce stand | Russia | 56.46 | 32.92 | ENF | Dfb | 1998-2010 | Kurbatova et al. (2008) |
| SE-Deg | Degero | Sweden | 64.18 | 19.55 | WET | Dfc | 2001-2009 | Sagerfors et al. (2008) |
| SE-Fla | Flakaliden | Sweden | 64.11 | 19.46 | ENF | Dfc | 1996-2002 | Valentini et al. (2000) |
| SE-Nor | Norunda | Sweden | 60.09 | 17.48 | ENF | Dfb | 1996-2007 | Lagergren et al. (2008) |
| US-Bo1 | IL - Bondville | USA | 40.01 | -88.29 | CRO | Dfa | 1996-2007 | Meyers and Hollinger (2004) |
| US-FPe | MT - Fort Peck | USA | 48.31 | -105.10 | GRA | BSk | 2000-2006 | Pataki and Oren (2003) |
| US-Ho1 | ME - Howland Forest (main tower) | USA | 45.20 | -68.74 | ENF | Dfb | 1996-2004 | Hollinger et al. (2004) |
| US-Ho2 | ME - Howland Forest (west tower) | USA | 45.21 | -68.75 | ENF | Dfb | 1999-2004 | Hollinger et al. (2004) |
| US-Ivo | AK - Ivotuk | USA | 68.49 | -155.75 | WET | ET | 2003-2006 | Epstein et al. (2004) |
| US-KS2 | FL - Kennedy Space Center | USA | 28.61 | -80.67 | CSH | Cfa | 2000-2006 | Powell et al. (2006) |
| US-MMS | IN - Morgan Monroe State Forest | USA | 39.32 | -86.41 | DBF | Cfa | 1999-2005 | Roman et al. (2015) |
| US-NR1 | CO - Niwot Ridge Forest | USA | 40.03 | -105.55 | ENF | Dfc | 1999-2003 | Monson et al. (2002) |
| US-SO4 | CA - Sky Oaks- New Stand | USA | 33.38 | -116.64 | CSH | Csa | 2004-2006 | Lipson et al. (2005) |
| US-SRM | AZ - Santa Rita Mesquite | USA | 31.82 | -110.87 | WSA | BSk | 2004-2006 | Scott et al. (2009) |
| US-Ton | CA - Tonzi Ranch | USA | 38.43 | -120.97 | WSA | Csa | 2001-2006 | Ma et al. (2007) |
| US-UMB | MI - Univ. of Mich. Biological Station | USA | 45.56 | -84.71 | DBF | Dfb | 1999-2003 | Gough et al. (2008) |
| US-Var | CA - Vaira Ranch- Ione | USA | 38.41 | -120.95 | GRA | Csa | 2001-2006 | Xu and Baldocchi (2004) |
| US-WCr | WI - Willow Creek | USA | 45.81 | -90.08 | DBF | Dfb | 1999-2006 | Cook et al. (2004) |
| US-Wrc | WA - Wind River Crane Site | USA | 45.82 | -121.95 | ENF | Csb | 1998-2006 | Falk et al. (2008) |

*Acknowledgements.* Authors affiliated with the MPI for Biogeochemistry acknowledge the European Union for funding via the H2020 project BACI (grant agreement No: 640176), This work used eddy covariance data acquired by the FLUXNET community and in particular by the following networks: AmeriFlux (U.S. Department of Energy, Biological and Environmental Research, Terrestrial Carbon Program

5  (DE-FG02-04ER63917 and DE-FG020-4ER63911)), AfriFlux, AsiaFlux, CarboAfrica, CarboEuropeIP, CarboItaly, CarboMont, ChinaFlux, Fluxnet-Canada (supported by CFCAS, NSERC, BIOCAP, Environment Canada, and NRCan), Canadian Carbon Program (supported by CFCAS, Environment Canada, and NRCan), GreenGrass, KoFlux, LBA, NECC, OzFlux, Swiss FluxNet, TCOS-Siberia, USCCC. We acknowledge the financial support to the eddy covariance data harmonization provided by CarboEuropeIP, FAO-GTOS-TCO, iLEAPS, Max Planck Institute for Biogeochemistry, National Science Foundation, University of Tuscia, Université Laval, Environment Canada and US

10  Department of Energy and the database development and technical support from Berkeley Water Center, Lawrence Berkeley National Laboratory, Microsoft Research eScience, Oak Ridge National Laboratory, University of California - Berkeley and the University of Virginia. Finally we thank Andrew Durso for helpful comments and assistance on the fine secrets of the English language.





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
