# Peer review of "Impacts of droughts and extreme temperature events on gross primary production and ecosystem respiration: a systematic assessment across ecosystems and climate zones"

_Biogeosciences, 2017_

## Referee Comment (RC1) · Anonymous Referee #1 · 16 Nov 2017

The authors investigated how climate extreme events altered GPP, Reco and NEE across global FLUXNET data sites. The key findings include that heatwave, drought, and heatwave*drought control GPP, Reco, and NEE differently, and the duration of the extreme events also control CO2 flux behaviors. Those findings have great relevance in global carbon cycle community, and will be useful for developing LSMs to better reflect climate extremes and C cycles. I have a few comments.

1. I am curious why the authors used LaThuile dataset which includes short term dataset than FLUXNET2015. This is particularly important as this manuscript deals

with climate extreme and C fluxes.

2. I am curious why Tmax related events are much more frequent than the others (Tmin, WAI..) in Fig 3a. The extremes were defined as 5

3. I recommend choosing a few (not all) long-term (>15 years) flux tower data from FLUXNET2015, and testing how your delta GPP, Reco, and NEE are robust with different time spans (e.g. 5, 10, 15 years) or random samples (say, 10 years) many times and check delta GPP, Reco and NEE. Although the authors started with Fig 1 stressing available long-term data, I feel many sites have <5 years data records, which might be not enough to test delta GPP, Reco, and NEE although they include climate extreme years. I think the authors already have all results for individual sites, so it would not require substantial efforts.

Specific comments: P8 L13: Curious why APAR was used in computing potential evaporation. Net radiation is a better proxy and is available from reanalysis datasets.

P9 L1: I think some ecosystems (e.g. savanna) reveal seasonally varying sensitivity to water availability (e.g. wet vs dry season).

P10 L27: remove ")"

P12 L7: add "("

P12 L17: Reco is computed using soil/air temperature from NEE. I am curious if such high sensitivity of Reco to temp extremes is entirely independent from the way to compute Reco.

P15 L8: The argument, "a stronger increase in Reco led more C gain" looks contradictory. Probably stronger increase in GPP?

P18 L27: add ")"

---

## Referee Comment (RC2) · Anonymous Referee #2 · 23 Nov 2017

General Comments

The manuscript is a relevant and very well written investigation of the role of extreme climatic events (temperature and precipitation) in determining GPP, Reco, and NEP across ecosystem types, based on global data from FLUXNET. Particularly valuable is the exploration of a simple standardized approach to identifying extreme events, which may result in inclusion of extreme events that might not be identified based on a priori identification of events. There are also potential pitfalls in this approach, which I believe the authors address reasonably well. In particular, the authors note that reliance on

percentiles may result in false positives for shorter time series and for sites with low climatic variability. The results are generally consistent with previous research, which is promising indication that the approach is valid. In any case, the automated approach will ...

Specific comments

Section 1.2 Provides thorough and very specific background on the response of photosynthesis to temperature and water stress (line 25 p. 3 to line 6 p. 4), but less detailed background on microbial responses to the same stressors (lines 7-14 p. 4). It would be great to balance this by providing a bit more background on microbial ecology, and perhaps streamline the background on photosynthetic responses.

Section 2.2 (line 5) is there a citation that could be added to justify the use of midday NEE as approximation for GPP and nighttime NEE for R_eco?

Section 2.3 (line 4 p.8) What is the relevance of noting that most sites have $R^2 < 0.9$? Is this a source of concern, or is the exclusion of sites with $R^2 < 0.6$ sufficient? Clarification would be useful.

Section 2.3 (line 10 p. 8) If i understand this description correctly, the authors are stating that their approach assumes that the water storage capacity is assumed the same everywhere. In absence of good site-specific information, this approach may be reasonable, but it would also be useful to provide some commentary on how the results might be affected by this assumption. For example, what would the sensitivity of the results be to assuming a slightly lower or higher water storage capacity?

Section 2.4 (p. 9 lines 2-3) Why is seasonal variation in sensitivity to water availability not expected? My intuition is that there could be quite a bit of sensitivity in ecosystems where phenology is driven by precipitation rather than temperature. Can the authors provide more explanation?

Section 3.3 p. 13 line 23 – This is a small thing, but having the longer term reversal

in the trend for R_eco described as "somewhat surprising" makes me wonder what evidence there is that we might expect any other trend. A little context on why the authors find it surprising would be helpful, or alternatively, I'd suggest just deleting the phrase "and somewhat surprisingly,".

---

## Author Comment (AC1) · 21 Dec 2017

We thank the reviewer for the positive and helpful comments. Please find below our detailed responses to their remarks (in italics):

*1. I am curious why the authors used LaThuile dataset which includes short term dataset than FLUXNET2015. This is particularly important as this manuscript deals with climate extreme and C fluxes.*

We understand that the reviewer is surprised to see that we work with the La Thuile

data set and this indeed needs justification: This activity started as a synthesis activity in the period where only the La Thuile dataset was available. However, please note that we already added more recent data for several sites available from the European flux network (http://www.europe-fluxdata.eu/) so we are basically using an extended La Thuile version of FLUXNET. Still, due to internal reasons, the paper could not be finalized earlier than 2017. We are also aware that the length of the time-series has increased in the US since we started the analysis and the record length of our analysis is not as long as it could have been. However, the complicated data harmonization, upscaling and WAI analyses with the La Thuile dataset were quite extensive and cannot be repeated because the first author has moved on to other duties. We therefore decided to progress with the publication using the La Thuile dataset. We see this paper as a prototype on how one could reanalyze the new FLUXNET releases but also as a benchmark for the results from the numerous papers published on the La Thuile version of the FLUXNET dataset.

*2. I am curious why Tmax related events are much more frequent than the others (Tmin, WAI..) in Fig 3a. The extremes were defined as 5*

First of all, this is due to the different durations of high temperature extremes detected compared to the detected WAI extremes. Due to the different data streams and climate variables used, temperature values tended to fluctuate with a much higher frequency compared to the rather smooth and slow fluctuations of WAI (cf. Fig 8 for an example). This resulted in many independent but rather short extreme events for temperature compared to fewer but longer extreme events for the strongly auto-correlated WAI. The amount of extreme **days** (in contrast to extreme **events**) was identical for both variables (5% of the data). The small amount of $T_{min}$ extreme events compared to $T_{max}$ is due to the fact that we only investigated (and, hence, plotted in Fig 3) instantaneous responses during the growing season (cf. Sec. 2.5). For extra tropical sites, many of the defined $T_{min}$ extreme events occurred during winter, which is outside the growing season. This explains the much lower number of extreme events in Fig 3.

*3. I recommend choosing a few (not all) long-term (>15 years) flux tower data from FLUXNET2015, and testing how your delta GPP, Reco, and NEE are robust with different time spans (e.g. 5, 10, 15 years) or random samples (say, 10 years) many times and check delta GPP, Reco and NEE. Although the authors started with Fig 1 stressing available long-term data, I feel many sites have <5 years data records, which might be not enough to test delta GPP, Reco, and NEE although they include climate extreme years. I think the authors already have all results for individual sites, so it would not require substantial efforts.*

First of all, we are not overly concerned about this issue, as the extremes were identified in the meteorological data streams which consisted of 30 year time series for every site. We then only compared the distributions of the flux anomalies during these extremes. Thus, longer flux time series would only make the estimated flux impacts slightly more robust but are unlikely to change the general results on extreme event distributions etc. discussed in this paper. In addition, due to changed professional locations, regrettably, we are unable to do this (see also our response to remark 1 above). We would have to redo the extensive analyses, including meteorological downscaling (which includes downloading current ERA interim datasets, harmonizing them, ,etc), new WAI calculations, etc..

*P8 L13: Curious why APAR was used in computing potential evaporation. Net radiation is a better proxy and is available from reanalysis datasets.*

Thanks for bringing this to our attention. In fact, net radiation is used in our estimation. We basically scaled potential evapotranspiration (which is estimated via the Priestley Taylor approach using net radiation) with fAPAR, which is a common approach. Our method description (and the more detailed description we are referring to in Tramontana et.al (2016)) does not, however, mention this explicitly. We extended our methods description (line 12ff, page 8) to make this more clear:

"Potential evapotranspiration is estimated based on Priestley and Taylor (1972) from

net radiation (also taken from the reanalysis data) using a Priestley-Taylor coefficient of 1.26. Potential evapotranspiration is then finally scaled with smoothed fAPAR (from MODIS)."

*P9 L1: I think some ecosystems (e.g. savanna) reveal seasonally varying sensitivity to water availability (e.g. wet vs dry season).*

The referee is right. See the corresponding answer to a similar remark from referee 2 (i.e. the 5th comment) for details.

*P10 L27: remove ")"*

We removed ")".

*P12 L7: add "("*

We added "(".

*P12 L17: Reco is computed using soil/air temperature from NEE. I am curious if such high sensitivity of Reco to temp extremes is entirely independent from the way to compute Reco.*

The reviewer raises very valid issue here. We are confident that this dependence of the different variables does not bias our results very much, as, using th approach of Reichstein et. al. (2005), the temperature dependence of Reco to NEE is computed using a rather short and local time window. If high temperatures during drought extremes would cause an impact on the ecosystem which would change this dependency, the flux partitioning algorithm would reflect this temporal change. We tested our confidence, however, by performing the identical analysis with night time NEE as a directly measured proxy for NEE and mid-day NEE as a proxy for GPP (page 7, line 2-6) and found patterns consistent with the results presented (data is not shown in the submitted manuscript).

*P15 L8: The argument, "a stronger increase in Reco led more C gain" looks contradic-*

*tory. Probably stronger increase in GPP?*

Thanks for bringing this to our attention, GPP and Reco were confused here. We changed this to:

"A stronger increase of GPP led to a slight overall increase of NEP (i.e. a C gain)"

*P18 L27: add ")"*

We added ")"

References

- Reichstein et al, (2005), Global Change Biology 11, 1–16, doi: 10.1111/j.1365-2486.2005.001002.x, On the separation of net ecosystem exchange into assimilation and ecosystem respiration: review and improved algorithm

- Tramontana, G., Jung, M., Schwalm, C. R., Ichii, K., Camps-Valls, G., Ráduly, B., Reichstein, M., Arain, M. A., Cescatti, A., Kiely, G., Merbold, L., Serrano-Ortiz, P., Sickert, S., Wolf, S., and Papale, D. (2016): Predicting carbon dioxide and energy fluxes across global FLUXNET sites with regression algorithms, Biogeosciences, 13, 4291-4313, https://doi.org/10.5194/bg-13-4291-2016.

---

## Author Comment (AC2) · 21 Dec 2017

We thank the reviewer for the positive and helpful comments. Please find below our detailed responses to their remarks (in italics):

*Section 1.2 Provides thorough and very specific background on the response of photo-synthesis to temperature and water stress (line 25 p. 3 to line 6 p. 4), but less detailed background on microbial responses to the same stressors (lines 7-14 p. 4). It would be great to balance this by providing a bit more background on microbial ecology, and*

*perhaps streamline the background on photosynthetic responses.*

We agree. We supplemented the discussion of extreme event impacts on ecosystem respiration with the following:

"Ecosystem respiration ($R_{eco}$) is the sum of autotrophic respiration and the CO2 emissions arising from the heterotrophic decomposition of organic matter in soil (e.g. Law et al., 1999, 2001; Epron et al., 2004). Like GPP, it is affected by changing soil (and, hence, ambient air) temperatures (Lloyd and Taylor, 1994; Kirschbaum, 1995; Davidson et al., 1998; Kirschbaum, 2006). Rising temperatures directly increase the kinetics of microbial decomposition, root respiration and the diffusion of enzymes. Hence, soil respiration is commonly modeled as an exponential function of temperature using the van't Hoff type Q10 model (van't Hoff 1898, Jassal 2008, Mahecha et.al. 2010) or other functions of similar shape (e.g. Kätterer et al., 1998, Kirschbaum, 1995 and Reichstein et.al, 2008). Even though enzyme activity generally decreases above a certain temperature optimum (Kirschbaum, 1995), such high temperatures rarely occur in extra tropical soils (Reichstein et.al., 2008), so high temperatures alone are rarely an inhibiting stressor for soil respiration.

In addition, the activity of soil microorganisms depends on soil moisture (Orchard and Cook, 1983; Gaumont-Guay et al., 2006; Liu et al., 2009; Epron et al., 2004). Drought conditions strongly reduce soil respiration because the microbial activity causing soil respiration is dependent on the presence of water films for substrate diffusion and exo-enzyme activity (Davidson & Janssens, 2006, Jassal et al., 2008, Frank et. al., 2015). In addition, low soil water status may even cause microbial dormancy and/or death (Orchard & Cook, 1983). Indirectly, drought reduces microbial activity through different processes like the alteration of soil nutrient retention and availability (Muhr et al., 2010; Bloor & Bardgett, 2012) or changes in microbial community structure (Sheik et al., 2011, Frank et.al. 2015). Finally, interactions between the response to temperature and water status, such as changing temperature dependency due to changing soil water status (e.g. Reichstein et al., 2002, 2007), further complicate the picture."

*Section 2.2 (line 5) is there a citation that could be added to justify the use of midday NEE as approximation for GPP and nighttime NEE for $R_{eco}$?*

Thanks for bringing this to our attention. We added a reference for this:

"To assess whether this could bias our analysis, we also performed all of our calculations using midday NEE as a rough estimate for GPP and averaged nighttime NEE as a proxy for $R_{eco}$ (Reichstein et. al. 2005)."

*Section 2.3 (line 4 p.8) What is the relevance of noting that most sites have $R^2$ <0.9? Is this a source of concern, or is the exclusion of sites with R2 < 0.6 sufficient? Clarification would be useful.*

We are sorry, but the "<" was a mere spelling mistake. We corrected it to ">":

"The correlation between downscaled and site-level data for air temperature was $R^2$ > 0.9 for nearly 90% of the sites."

*Section 2.3 (line 10 p. 8) If i understand this description correctly, the authors are stating that their approach assumes that the water storage capacity is assumed the same everywhere. In absence of good site-specific information, this approach may be reasonable, but it would also be useful to provide some commentary on how the results might be affected by this assumption. For example, what would the sensitivity of the results be to assuming a slightly lower or higher water storage capacity?*

The reviewer raises a very valid issue here. The water storage capacity is highly dependent on various soil specifics. However, to consistently calculate a water status estimator for the wide range of sites investigated here without the availability of consistent and comparable water storage capacity data for all sites, we had to use this rather crude simplification. We added the following paragraph to the methodological description of WAI (page 8, line 18):

"The WAI does not account for local soil or vegetation specific properties such as soil texture, rooting depth, etc. such that the WAI may be interpreted as a 'climatological

water availability metric'. The results are sensitive to the fixed value of storage capacity, which influences the timing and magnitude of extreme drought events. For example, a larger (smaller) storage capacity value would tend to result in a later (earlier) extreme drought detection. We are confident, however, that these changes would not strongly bias the qualitative and global patterns of the flux impacts investigated in this analysis."

*Section 2.4 (p. 9 lines 2-3) Why is seasonal variation in sensitivity to water availability not expected? My intuition is that there could be quite a bit of sensitivity in ecosystems where phenology is driven by precipitation rather than temperature. Can the authors provide more explanation?*

We agree, for several ecosystems the response to WAI can certainly vary seasonally. Simply de-seasonalizing the WAI time series (like we did with temperature) would, however, not be a good solution to this. To fully address this issue, it would be necessary to systematically stratify the extreme events according to the phenological stages or timing during the growing season. This was, however, not the scope of this first assessment, but may be the focus of future studies using the presented framework (see page 22, line 27).

*Section 3.3 p. 13 line 23 – This is a small thing, but having the longer term reversal in the trend for $R_{eco}$ described as "somewhat surprising" makes me wonder what evidence there is that we might expect any other trend. A little context on why the authors find it surprising would be helpful, or alternatively, I'd suggest just deleting the phrase "and somewhat surprisingly,"*

The referee is right, it is not that "surprising" that the negative impacts on $R_{eco}$ increased with duration. With our formulation we wanted to stress the importance of the fact that short extremes generally showed increasing effects on $R_{eco}$, whereas longer extremes resulted in mostly decreases in $R_{eco}$. We changed the paragraph as follows to clarify this aspect:

"With increasing duration, the response in $R_{eco}$ was reversed: for short heat events (i.e.

a duration of less than 18 days), $R_{eco}$ increased with respect to normal conditions by up to 2 $\sigma$, whereas for events that last longer than a month, the response of $R_{eco}$ was predominantly negative."

References:

(in addition to the ones already cited in the discussion version of the manuscript):

- Bloor JMG, Bardgett RD, 2012, Stability of above-ground and below-ground processes to extreme drought in model grassland ecosystems: Interactions with plant species diversity and soil nitrogen availability. Perspectives in Plant Ecology Evolution and Systematics, 14, 193- 204.

- Davidson EA, Janssens IA, 2006, Temperature sensitivity of soil carbon decomposition and feedbacks to climate change. Nature, 440, 165-173.

- Kätterer, T., Reichstein, M., Andrén, O., Lomander, A. (1998): Temperature dependence of organic matter decomposition: a critical review using literature data analyzed with different models. Biol. Fertil. Soils 27, 258–262.

- Miguel D. Mahecha, Markus Reichstein, Nuno Carvalhais, Gitta Lasslop, Holger Lange, Sonia I. Seneviratne, Rodrigo Vargas, Christof Ammann, M. Altaf Arain, Alessandro Cescatti, Ivan A. Janssens, Mirco Migliavacca, Leonardo Montagnani, Andrew D. Richardson (2010). Global convergence in the temperature sensitivity of respiration at ecosystem level. Science, 329(5993), 838-840.

- Muhr J, Franke J, Borken W (2010) Drying-rewetting events reduce C and N losses from Norway spruce forest floor. Soil Biology & Biochemistry, 42, 1303-1312

- Reichstein et al, (2005), Global Change Biology 11, 1–16, doi: 10.1111/j.1365-2486.2005.001002.x, On the separation of net ecosystem exchange into assimilation and ecosystem respiration: review and improved algorithm

- Reichstein, M., & Beer, C. (2008). Soil respiration across scales: the importance of a model–data integration framework for data interpretation. Journal of Plant Nutrition and Soil Science, 171(3), 344-354.

- Sheik CS, Beasley WH, Elshahed MS, Zhou X, Luo Y, Krumholz LR (2011) Effect of warming and drought on grassland microbial communities. The ISME Journal - Multidisciplinary Journal of Microbial Ecology, 5, 1692-1700.

- van't Hoff, J. H. (1898): Lectures on Theoretical and Physical Chemistry. Part 1: Chemical Dynamics. Edward Arnold, London